# QUIC-FL: QUICK UNBIASED COMPRESSION FOR FEDERATED LEARNING

## ABSTRACT

Distributed Mean Estimation (DME) is a fundamental building block in communication efficient federated learning. In DME, clients communicate their lossily compressed gradients to the parameter server, which estimates the average and updates the model. State of the art DME techniques apply either unbiased quantization methods, resulting in large estimation errors, or biased quantization methods, where unbiasing the result requires that the server decodes each gradient individually, which markedly slows the aggregation time. In this paper, we propose QUIC-FL, a DME algorithm that achieves the best of all worlds. QUIC-FL is unbiased, offers fast aggregation time, and is competitive with the most accurate (slow aggregation) DME techniques. To achieve this, we formalize the problem in a novel way that allows us to use standard solvers to design near-optimal unbiased quantization schemes.

## 1  INTRODUCTION

In federated learning McMahan et al. (2017); Kairouz et al. (2019), clients periodically send their gradients to the parameter server, which calculates their means. This communication is often a network bottleneck, and methods to approximate the mean using small communication are desirable. The Distributed Mean Estimation problem (DME) Suresh et al. (2017) formalizes this fundamental building block as follows: each of $n$ clients communicate a representation of a $d$-dimensional vector to a *parameter server* which estimates the vectors' mean.

Various DME methods have been studied (e.g., Suresh et al. (2017); Konečný & Richtárik (2018); Vargaftik et al. (2021); Davies et al. (2021); Vargaftik et al. (2022)), examining tradeoffs between the required bandwidth and performance metrics such as the estimation accuracy, learning speed, and the eventual accuracy of the model.

These works utilize lossy compression techniques, using only a small number of bits per coordinate, which is shown to accelerate the training process Bai et al. (2021); Zhong et al. (2021). For example, in Suresh et al. (2017), each client randomly rotates its vector before applying stochastic quantization. When receiving the messages from the clients, the server sums up the estimates of the rotated vectors and applies the inverse rotation. As the largest coordinates are asymptotically larger than the mean, their Normalized Mean Squared Error ($NMSE$) is bounded by $O\left(\log d/n\right)$. They also propose an entropy encoding method that reduces the $NMSE$ to $O(1/n)$ but is slow and not GPU-friendly. A different approach to DME computes the Kashin's representation Lyubarskii & Vershynin (2010) of a client's vector before applying quantization Caldas et al. (2018); Safaryan et al. (2020). Intuitively, this replaces the input $d$-dimensional vector by $\lambda \cdot d$ coefficients, for some $\lambda > 1$, each bounded by $O\left(\sqrt{\|x\|_2}/d\right)$. Applying quantization to the coefficients instead of the original vectors allows the server to estimate the mean using $\lambda > 1$ bits per coordinate with an $NMSE$ of $O\left(\frac{\lambda^2}{(\sqrt{\lambda}-1)^4 \cdot n}\right)$. However, it requires applying multiple randomized Hadamard transforms, slowing down its encoding.

The recently introduced DRIVE Vargaftik et al. (2021) (which uses $b = 1$ bits per coordinate) and its generalization EDEN Vargaftik et al. (2022) (that works with any $b > 0$) also randomly rotate the input vector, but unlike Suresh et al. (2017) use *biased* (deterministic) quantization on the rotated coordinates. Interestingly, both yield *unbiased* estimates of the input vector after multiplying the estimated vector by a real-valued "scale" that is sent by each client together with the quantization. Both solutions have an $NMSE$ of $O(1/n)$ and are empirically more accurate than Kashin's representation. However, to achieve unbiasedness, each client must generate a distinct rotation matrix independently

| Algorithm | Enc. complexity | Dec. complexity | NMSE |
|---|---|---|---|
| **QSGD** Alistarh et al. (2017) | $O(d)$ | $O(n \cdot d)$ | $O(d/n)$ |
| **Hadamard** Suresh et al. (2017) | $O(d \cdot \log d)$ | $O(n \cdot d + d \cdot \log d)$ | $O(\log d/n)$ |
| **Kashin** Caldas et al. (2018); Safaryan et al. (2020) | $O(d \cdot \log d \cdot \log(n \cdot d))$ | $O(n \cdot d + d \cdot \log d)$ | $O(1/n)$ |
| **EDEN** Vargaftik et al. (2022) | $O(d \cdot \log d)$ | $O(n \cdot d \cdot \log d)$ | $O(1/n)$ |
| **QUIC-FL (New)** | $O(d \cdot \log d)$ | $O(n \cdot d + d \cdot \log d)$ | $O(1/n)$ |

**Table 1:** The asymptotic guarantees of the algorithms with $b = O(1)$ bits per coordinate and using the Hadamard transform for rotation based algorithms. The table does not consider variable length encodings (see Appendix A).

from other clients. In turn, the server must invert the rotation for each vector before aggregating them, resulting in $O(n)$ rotations instead of one, asymptotically increasing the decoding time.

Here we attempt to resolve the decoding time slowdown from these recent state of the art DME techniques Vargaftik et al. (2021; 2022). Again, this slowdown arises because unbiasing the estimates requires each client must use its own *independent* random rotation, and accordingly the server must invert the rotation for each quantized gradient.

In this work we present **Q**uick **U**nb**i**ased **C**ompression for **F**ederated **L**earning (QUIC-FL): a DME algorithm that produces unbiased estimates, with a fast estimation procedure and an NMSE of $O(1/n)$. QUIC-FL also leverages random rotations, and uses the observation that after rotation the coordinates' distribution approaches $d$ i.i.d. normal variables, $\mathcal{N}(0, \|x\|_2/d)$ Vargaftik et al. (2021). The goal of QUIC-FL is to unbiasedly quantize each coordinate while minimizing the error. Compared with Suresh et al. (2017), we present two key improvements: (1) Instead of quantizing all coordinates, we allow the algorithm to send an expected $p$-fraction of the *rotated* coordinates exactly (up to precision) for some small $p$ (e.g., $p = 1/512$). This limits the range of the other coordinates to $[-T_p, T_p]$, where $T_p = O(1)$ for any constant $p > 0$, thus reducing the possible quantization error significantly. (2) We study how to leverage *client-specific shared randomness* Ben Basat et al. (2021) to reduce the error further. Specifically, we model the problem of transmitting a "bounded-support" normal random variable $Z \sim \mathcal{N}(0,1) \mid Z \in [-T_p, T_p]$, using $b \in \mathbb{N}^+$ bits, with the goal of obtaining an unbiased estimate at the server. Our model considers both a client's private randomness and shared randomness between the clients and server, allowing us to derive an input to optimization problem solver, whose output yields algorithms with a near-optimal accuracy to bandwidth tradeoff.

We implement QUIC-FL in PyTorch Paszke et al. (2019) and TensorFlow Abadi et al. (2015), showing that it can compress vectors with over 33 million coordinates within 44 milliseconds and is markedly more accurate than existing fast-estimate approaches such as QSGD Alistarh et al. (2017), Hadamard Suresh et al. (2017), and Kashin Caldas et al. (2018); Safaryan et al. (2020). Compared with DRIVE Vargaftik et al. (2021) and EDEN Vargaftik et al. (2022), QUIC-FL has only slightly worse NMSE (e.g., less than 1% for $b = 4$ bits per

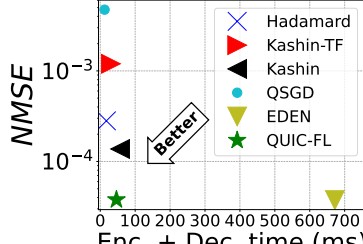

dimension) while asymptotically improving the estimation time, as shown on the right. The figure illustrates the cycle (encode plus decode) times vs. NMSE for $b = 4$ bits per coordinate, $d = 2^{20}$ dimensions, and $n = 256$ clients. (see §4 for the algorithms' description.) We summarize the asymptotic guarantees of the discussed DME techniques in Table 1.

We note that our algorithm is based on deriving near-optimal stochastic quantizations for a specific distribution by determining a mathematical program (a set of constraints) fed to an optimization program solver. We believe this approach will prove useful for other problems that use stochastic quantization.

While we have surveyed the most relevant related work above, we review other techniques in Appendix A. (All appendices appear in the supplementary material.)

## 2 PRELIMINARIES

**Problems and Metrics.** Given a non-zero vector $x \in \mathbb{R}^d$, a vector compression protocol consists of a client that computes a message $X$ and a server that given the message estimates $\hat{x} \in \mathbb{R}^d$. The *vector Normalized Mean Squared Error* (*vNMSE*) of the protocol is defined as $\frac{\mathbb{E}[\|x - \hat{x}\|_2^2]}{\|x\|_2^2}$ Vargaftik et al. (2021; 2022).

This problem generalizes to the Distributed Mean Estimation (DME) problem, where $n$ clients have vectors $\{x_c \in \mathbb{R}^d\}$ that they communicate to a centralized server. We are interested in minimizing the *Normalized Mean Squared Error* (*NMSE*), defined as $\frac{\mathbb{E}\left[\left\|\widehat{x}_{avg} - \frac{1}{n}\sum_{c=1}^n x_c\right\|_2^2\right]}{\frac{1}{n}\cdot\sum_{c=1}^n \|x_c\|_2^2}$ Suresh et al. (2017); Vargaftik et al. (2021; 2022), where $\widehat{x}_{avg}$ is our estimate of the average $\frac{1}{n}\sum_{c=1}^n x_c$. Note that for unbiased algorithms and independent estimates, we have that $NMSE = vNMSE/n$ Vargaftik et al. (2021).

**Shared randomness.** We use both global (common to all clients and the server) and client-specific shared randomness (one client and the server). Client-only randomness is termed *private* randomness.

## 3 THE QUIC-FL ALGORITHM

### 3.1 BOUNDED-SUPPORT-QUANTIZATION

Our first contribution is the introduction of bounded-support-quantization (BSQ). For a parameter $p \in (0, 1]$, we pick a threshold $T_p$ such that at most $d \cdot p$ coordinates can fall outside $[-T_p, T_p]$. BSQ separates the vector into two parts: the large coordinates whose absolute value is at least $T_p$, and the small ones. The large values are sent exactly (matching the precision of the input gradient), whereas the small values are quantized and transmitted using a small number of bits each.

This simple approach decreases the error of every quantized coordinate by bounding the quantized coordinates' support at the cost of transmitting some entries accurately. As stated in Appendix G, we formally show that BSQ, without further assumptions, admits a worst-case $vNMSE$ of $\frac{1}{p \cdot (2^b - 1)^2}$.

In particular, when $p$ and $b$ are constants, we get an $NMSE$ of $O(1/n)$ with encoding and decoding times of $O(d)$ and $O(nd)$, respectively. However, the linear dependence on $p$ means that the hidden constant in the $O(1/n)$ $NMSE$ is too large to be practical. For example, if $p = 2^{-5}$ and $b = 1$, we need two bits per coordinate on average: one for sending the exact values (assuming coordinates are single precision floats) and another for stochastically quantizing the remaining coordinates. In turn, we get a $vNMSE$ bound of $\frac{1}{2^{-5} \cdot (2^1 - 1)^2} = 32$. In the following section, we show that combining BSQ with random rotation allows us to get an $O(1/n)$ $NMSE$ even with a low constant for low values of $p$. For example, $p = 2^{-9}$ and an additional one bit per coordinate for the quantization, we reach a $vNMSE$ of 1.52, a $21\times$ improvement despite using less bandwidth.

### 3.2 ROTATIONS WITH BOUNDED SUPPORT QUANTIZATION

Similarly to previous works Suresh et al. (2017); Vargaftik et al. (2021; 2022), our algorithm QUIC-FL begins by randomly rotating the input vector, after which the coordinates' distribution approaches independent normal random variables for high dimensions Vargaftik et al. (2021). This effectively turns every input into the average case. We note that, unlike Vargaftik et al. (2021; 2022) all clients use the same rotation, generated with global shared randomness. QUIC-FL then utilizes near-optimal *unbiased* quantization for the normal distribution for each coordinate. We emphasize that QUIC-FL is unbiased for any input; the quantization is tuned for the normal distribution, as after rotation each coordinate it is well-approximated by a normal distribution.[1] Unlike previous algorithms, we combine the rotation with bounded support quantization. QUIC-FL achieves unbiasedness using both private randomness at the client and client-specific shared randomness (shared between it and the server).

As another comparison point, Suresh et al. (2017), given a bit budget of $b(1 + o(1))$ bits per packet, stochastically quantizes each rotated coordinate into one of $2^b$ levels. The algorithm uses a max-min normalization, and the levels are uniformly spaced between the minimal and maximal coordinates. Their algorithm then communicates the max and min, together with $b$ bits per coordinate indicating its quantized level, and is shown to have a $NMSE$ of $O\left(\log d/n\right)$ for any $b = O(1)$.

We begin by analyzing the value of rotation with BSQ. Let $Z = \mathcal{N}(0, 1)$ be a normal random variable, modeling a rotated (and scaled) coordinate. Given a user-defined parameter $p$, we can compute a

---

[1]For any fixed dimension, we can optimize the quantization (using the solver) for its exact rotated coordinate distribution (known to be a shifted-beta). However, this would require a different quantization for each dimension. The difference using a normal distribution is negligible, even with dimensions in the hundreds.

threshold $T_p$ such that $\Pr\left[Z \notin \left[-T_p, T_p\right]\right] = p$. For example, by picking $p = 2^{-9}$ (i.e., less than 0.2%), we get a threshold of $T_p \approx 3.097$.[2]

In general, for any *constant* $p > 0$, we have $T_p$ is constant, and using $b$ bits for each coordinate in $\left[-T_p, T_p\right]$ we get a *NMSE* of $O\left(1/n\right)$ for any constant $b$ (due to unbiased and independent quantization among clients). For example, consider sending each coordinate in $\left[-T_p, T_p\right]$ using $b = 1$ bit per coordinate. One solution would be to use stochastic quantization, i.e., given a coordinate $Z \in \left[-T_p, T_p\right]$ send the bit for which $\widehat{Z} = T_p$ with probability $\frac{Z+T_p}{2T_p}$ and the bit for $\widehat{Z} = -T_p$ otherwise. This quantization results in an unbiased estimate with expected squared error of

$$\mathbb{E}\left[(Z - \widehat{Z})^2\right] = \frac{1}{\sqrt{2\pi}}\int_{-T_p}^{T_p}\left(\frac{z+T_p}{2T_p}\cdot(z-T_p)^2 + \frac{T_p-z}{2T_p}\cdot(z+T_p)^2\right)\cdot e^{-\frac{z^2}{2}}\,dz.$$

With $p = 2^{-9}$ as above, we get $\mathbb{E}\left[(Z-\widehat{Z})^2\right] \approx 8.58$. We can view the algorithm expressed so far as a special case of QUIC-FL without shared randomness.

As shown in Appendix B, for QUIC-FL (with or without shared randomness) on any $d$-dimensional input vector (and any quantization scheme for $Z \in \left[-T_p, T_p\right]$), $vNMSE = \mathbb{E}\left[\left(Z-\widehat{Z}\right)^2\right] + O\left(\sqrt{\frac{\log d}{d}}\right)$. The additional additive term occurs because we chose to optimize for the normal distribution[3] Again, this holds for *any* initial vector because QUIC-FL starts with a random rotation. Thus, using the above quantization for each coordinate for large gradients results in $NMSE \approx 8.58/n$. We next show that additionally using client-specific shared randomness can decrease $\mathbb{E}[(Z - \widehat{Z})^2]$ and thus the *NMSE*.

### 3.3 Leveraging Client-specific Shared Randomness

We now provide an example to show how shared randomness can improve the *vNMSE*, leading to §3.4 where we formalize our approach to finding near-optimal unbiased compression schemes for bounded-support $\mathcal{N}(0,1)$ variables. Using a single shared random bit (i.e., $H \in \{0,1\}$), we can use the following algorithm, where $X$ is the sent message and $\alpha = 0.8, \beta = 5.4$ are constants:

$$X = \begin{cases} 1 & \text{if } H = 0 \text{ and } Z \geqslant 0 \\ 0 & \text{if } H = 1 \text{ and } Z < 0 \\ Bernoulli(\frac{2Z}{\alpha+\beta}) & \text{If } H = 1 \text{ and } Z \geqslant 0 \\ 1 - Bernoulli(\frac{-2Z}{\alpha+\beta}) & \text{If } H = 0 \text{ and } Z < 0 \end{cases} \qquad \widehat{Z} = \begin{cases} -\beta & \text{if } H = X = 0 \\ -\alpha & \text{if } H = 1 \text{ and } X = 0 \\ \alpha & \text{If } H = 0 \text{ and } X = 1 \\ \beta & \text{If } H = X = 1 \end{cases}.$$

For example, if $Z = 1$, then with probability $1/2$ we have that $H = 0$ and thus $X = 1$, and otherwise the client sends $X = 1$ with probability $\frac{2}{\alpha+\beta}$ (and otherwise $X = 0$). Similarly, the reconstruction would be $\widehat{Z} = \alpha$ with probability $1/2$ (when $H = 0$), $\widehat{Z} = \beta$ with probability $1/2 \cdot \frac{2}{\alpha+\beta} = 0.16$, and $\widehat{Z} = -\alpha$ with probability $1/2 \cdot \frac{\alpha+\beta-2}{\alpha+\beta} = 0.84$. Indeed, we have that the estimate is unbiased since:

$$\mathbb{E}[\widehat{Z} \mid Z = 1] = \alpha \cdot 1/2 + \beta \cdot 1/2 \cdot \frac{2}{\alpha+\beta} + (-\alpha) \cdot 1/2 \cdot \frac{\alpha+\beta-2}{\alpha+\beta} = 1.$$

We calculate the quantization's expected squared error, conditioned on $Z \in \left[-T_p, T_p\right]$. (From symmetry, we integrate over positive $t$.)

$$\mathbb{E}\left[(Z - \widehat{Z})^2\right] = \sqrt{\frac{2}{\pi}}\left(\int_0^{T_p}\frac{1}{2}\cdot\left((z-\alpha)^2 + \frac{2z}{\alpha+\beta}\cdot(z-\beta)^2 + \frac{\alpha+\beta-2z}{\alpha+\beta}\cdot(z+\alpha)^2\right)\cdot e^{-z^2/2}dz\right)$$

Using the same $p = 2^{-9}$ parameter ($T_p \approx 3.097$), we get an error of $\mathbb{E}\left[(Z - \widehat{Z})^2\right] \approx 3.29$, 61% lower than without shared randomness. This algorithm is derived from the solver, which numerically approximates the optimal unbiased algorithm with a single shared random bit, in terms of expected squared error, for this $p$. We present our general approach for using the solver in the following sections.

---

[2]Note that, as we begin with a normal random variable $Z$, bounding its support is effective at removing the long, small-probability tails. Additionally, as the learning process typically uses 16-64 bit floats, and we further need to send the coordinate indices, sending each coordinate is expensive, and thus we focus on small $p$ values.

[3]If instead we were quantizing a shifted-beta random variable $B$, we would get $vNMSE = \mathbb{E}\left[B - \widehat{B}\right]$.

### 3.4 Designing Near-optimal Unbiased Compression Schemes

In order to design our post-rotation compression scheme, we first model the problem as follows:

- We first choose a parameter $p > 0$, the expected fraction of coordinates allowed to be sent exactly.
- The input, known to the client, is a coordinate $Z \sim \mathcal{N}(0, 1)$. The $p$ parameter restricts further the distribution to $Z \in [-T_p, T_p]$.
- The client-specific shared randomness $H$ is known to both the client and server, and without loss of generality, we assume that $H \sim U[0, 1]$. We denote by $\mathcal{H} = [0, 1]$ the domain of $H$.
- We use a bit budget of $b \in \mathbb{N}^+$ bits per coordinate, and accordingly assume that the messages are in the set $\mathcal{X}_b = \{0, \ldots, 2^b - 1\}$.[4] Again, coordinates outside the range $[-T_p, T_p]$ are sent exactly.
- The client is modeled as $S : \mathcal{H} \times \mathbb{R} \to \Delta(\mathcal{X}_b)$. That is, the client observes the shared randomness $H$ and the input $Z$, and chooses a distribution over the messages. We further denote by $S_x(h, z)$ the probability that the client sends $x \in \mathcal{X}_b$ given $h$ and $z$ (i.e., $\forall h, z : \sum_x S_x(h, z) = 1$). For example, it may choose $S_x(0, 0) = \begin{cases} 1/2 & \text{If } x \in \{0, 1\} \\ 0 & \text{Otherwise} \end{cases}$. That is, given $z = h = 0$, the client shall use private randomness to decide whether to send $x = 0$ or $x = 1$, each with probability $1/2$.
- The server is modeled as a function $R : \mathcal{H} \times \mathcal{X}_b \to \mathbb{R}$, such that if the shared randomness is $h \in \mathcal{H}$ and the server receives the message $x \in \mathcal{X}_b$, it produces an estimate $\hat{z} = R(h, x)$.
- We require that the estimates are unbiased, i.e., $\mathbb{E}[\hat{Z} \mid Z] = Z$, where the expectation is taken over both the client-specific shared randomness $H$ and the private randomness of the client.

We are now ready to formally define the optimal unbiased quantization problem:

$$\underset{S,R}{\text{minimize}} \quad \frac{1}{\sqrt{2\pi}} \int_{-T_p}^{T_p} \int_0^1 \sum_x S_x(h, z) \cdot (z - R(h, x))^2 \cdot e^{-z^2/2} \, dh \, dz$$

$$\text{subject to} \quad \int_0^1 \sum_x S_x(h, z) \cdot R(h, x) \, dh = z, \qquad \forall z \in [-T_p, T_p].$$

We are unaware of methods for solving the above problem analytically. Instead, we propose a discrete relaxation of the problem, allowing us to approach it with a *solver*.[5] Namely, we model the algorithm as an optimization problem and let the solver output the optimal algorithm. To that end, we need to discretize the problem. Specifically, we make the following relaxations:

- The shared randomness $H$ is selected uniformly at random from a finite set of values $\mathcal{H}_\ell \triangleq \{0, \ldots, 2^\ell - 1\}$, i.e., using $\ell$ shared random bits.
- The bounded-support distribution of a rotated and scaled $Z \sim \mathcal{N}(0, 1)$ coordinate is approximated using a finite set of *quantiles* $\mathcal{Q}_m = \{q_0, \ldots, q_{m-1}\}$, for a parameter $m \in \mathbb{N}^+$. In particular, the quantile $q_i$ is the point on the CDF of the bounded-support normal distribution (restricted to $[-T_p, T_p]$) such that the $\Pr[Z \leqslant q_i \mid Z \in [-T_p, T_p]] = \frac{i}{m-1}$. Notice that we have $m$ such quantiles, corresponding to the probabilities $\left\{0, \frac{1}{m-1}, \frac{2}{m-1}, \ldots, 1\right\}$. For example, $p = 2^{-9}$ and $m = 4$ we get the quantile set $\mathcal{Q}_4 \approx \{-3.097, -0.4298, 0.4298, 3.097\}$.
- The client is now modeled as $S : \mathcal{H}_\ell \times \mathcal{Q}_m \to \Delta(\mathcal{X}_b)$. That is, for each shared randomness $h \in \mathcal{H}_\ell$ and quantile $q \in \mathcal{Q}_m$ values, the client has a *probability distribution* on the messages from which it samples, using private randomness, at encoding time.
- The server is modeled as a function $R : \mathcal{H}_\ell \times \mathcal{X}_b \to \mathbb{R}$, such that if the shared randomness is $H$ and the server receives the message $X$, it produces an estimate $\hat{Z} = R(H, X)$.

Given this modeling, we use the following variables:

---

[4]We note that using entropy encoding, one may use more than $2^b$ messages (and thereby reduce the error) if the resulting entropy is bounded by $b$ (e.g., Suresh et al. (2017); Vargaftik et al. (2022); Alistarh et al. (2017)). As we aim to design a quick and GPU-friendly compression scheme, we do not investigate entropy encoding further.

[5]We used the Gekko Beal et al. (2018) software package that provides a Python wrapper to the APMonitor Hedengren et al. (2014) environment, running the solvers IPOPT IPO and APOPT APO .

- $s = \{s_{h,q,x} \mid h \in \mathcal{H}_\ell, \ q \in \mathcal{Q}_m, \ x \in \mathcal{X}_b\}$, where $s_{h,q,x}$ denotes the probability of sending a message $x$, given the quantile $q$ and shared randomness value $h$. We note that the solver's solution will only instruct us what to do if all our coordinates were quantiles in $\mathcal{Q}_m$. In what follows, we show how to interpolate the result and get a practical algorithm for any $Z \in [-T_p, T_p]$.
- $r = \{r_{h,x} \mid h \in \mathcal{H}_\ell, \ x \in \mathcal{X}_b\}$, where $r_{h,x}$ denotes the server's estimate value given the shared randomness $h$ and the received message $x$.

Accordingly, the discretized unbiased quantization problem is defined as:

$$
\begin{aligned}
\underset{s,r}{\text{minimize}} \quad & \frac{1}{m} \cdot \frac{1}{2^\ell} \cdot \sum_{h,q,x} s_{h,q,x} \cdot (q - r_{h,x})^2 \\
\text{subject to} \quad & \\
(\textit{Unbiasedness}) \quad & \frac{1}{2^\ell} \cdot \sum_{h,x} s_{h,q,x} \cdot r_{h,x} = q, \qquad & \forall q \\
(\textit{Probability}) \quad & \sum_x s_{h,q,x} = 1, \qquad & \forall h, q \\
& s_{h,q,x} \geqslant 0, \qquad & \forall h, q, x
\end{aligned}
$$

As mentioned, the solver's output does not directly yield an implementable algorithm, as it only associates probabilities to each $\langle h, q, x \rangle$ tuple. A natural option is to first stochastically quantize $Z$ to a quantile. For example, when $Z = 1$ and using the $\mathcal{Q}_4$ described above, before applying the algorithm, we quantize it to $q^- = 0.4298$ with probability $\approx 0.786$ or $q^+ = 3.097$ with probability $\approx 0.214$.

This approach gives an algorithm whose pseudo-code is given in Algorithm 1. The resulting algorithm is near-optimal in the sense that as the number of quantiles and shared random bits tend to infinity, we converge to an optimal algorithm. In practice, the solver is only able to produce an output for finite $m, \ell$ values; this means that the algorithm would be optimal if coordinates are uniformly distributed over $\mathcal{Q}_m$, and not in $\mathcal{N}(0, 1)$.

In words, in Algorithm 1 each client $c$ uses shared randomness to compute a global random rotation $\mathcal{R}$ (note that all clients use the same rotation). Next, it computes the rotated vector $\mathcal{R}(x_c)$; for sufficiently large dimensions, the distribution of each entry in $\overline{Z}_c$ converges to $\mathcal{N}\left(0, \frac{\|x_c\|_2^2}{d}\right)$. The client then normalizes it, $\overline{Z}_c = \frac{\sqrt{d}}{\|x_c\|_2} \cdot \mathcal{R}(x_c)$, to have the coordinates roughly distributed $\mathcal{N}(0, 1)$. Next, it stochastically quantizes the vector to $\mathcal{Q}_m$. Namely, for a given coordinate $Z$, let $q^-, q^+ \in \mathcal{Q}_m$ denote the largest quantile smaller or equal to $Z$, and the smallest quantile larger than $q$ respectively. Then we denote by $\mathcal{Q}_m(Z)$ the stochastic quantization operation that returns $q^+$ with probability $\frac{Z - q^-}{q^+ - q^-}$ and $q^-$ otherwise. The stochastic quantization of the vector applies coordinate-wise, i.e., $\mathcal{Q}_m(\overline{Z}_c) = (\mathcal{Q}_m(\overline{Z}_c[0]), \ldots, \mathcal{Q}_m(\overline{Z}_c[d-1]))$. The next step is to generate a client-specific shared randomness vector $\overline{H}_c$ in which each entry is drawn uniformly and independently from $\mathcal{H}_\ell$. Finally, the client follows the client algorithm produced by the solver. That is, for each coordinate $Z$, the client takes the mapped quantile $q = \mathcal{Q}_m(Z) \in \mathcal{Q}_m$, considers the set of probabilities $\{s_{h,q,x} \mid x \in \mathcal{X}_b\}$, and samples a message accordingly. We denote applying this operation coordinate-wise by $\overline{X}_c \sim \left\{x \text{ with prob. } s_{\overline{H}_c, \tilde{Z}_c, x} \mid x \in \mathcal{X}_b\right\}$. It then sends the resulting vector $\overline{X}_c$ to the server, together with the norm $\|x_c\|_2$. In turn, for each client $c$, the server estimates its rotated vector by looking up the shared randomness and message for each coordinate. That is, given $\overline{H}_c = (\overline{H}_c[0], \ldots, \overline{H}_c[d-1])$ and $\overline{X}_c = (\overline{X}_c[0], \ldots, \overline{X}_c[d-1])$ we denote $r_{\overline{H}_c, \overline{X}_c} = (r_{\overline{H}_c[0], \overline{X}_c[0]}, \ldots)$. The server then estimates $\mathcal{R}(x_c)$ as $\left(\|x_c\| / \sqrt{d} \cdot r_{\overline{H}_c, \overline{X}_c}\right)$ and averages across all clients before performing the inverse rotation. In the next section, we analyze the solver's output and show how to improve this method.

**Further optimization**  A different approach to yield an implementable algorithm from the optimal solution to the discrete problem is to calculate the message distribution directly from the rotated values without stochastically quantizing as we do in Line 2. Indeed, we have found this approach to be somewhat faster and more accurate. Due to space constraints, we defer the details to Appendix C.

---

**Algorithm 1**

---

**Client $c$:**
1: Compute $\overline{Z}_c = \frac{\sqrt{d}}{\|x_c\|_2} \cdot \mathcal{R}(x_c)$.
2: Stochastically quantize $\widetilde{Z}_c = \mathcal{Q}_m(\overline{Z}_c)$
3: Sample $\overline{X}_c \sim \left\{ x \text{ with prob. } s_{\overline{H}_c, \tilde{z}_c, x} \mid x \in \mathcal{X}_b \right\}$
4: Send $\left( \|x_c\|_2, \overline{X}_c \right)$ to server

**Server:**
1: $\forall c$ : Compute $\widehat{\overline{Z}}_c = r_{\overline{H}_c, \overline{X}_c}$
2: Compute $\widehat{\overline{Z}}_{avg} = \frac{1}{n} \cdot \frac{1}{\sqrt{d}} \cdot \sum_{c=1}^n \|x_c\|_2 \cdot \widehat{\overline{Z}}_c$
3: Estimate $\hat{x}_{avg} = \mathcal{R}^{-1} \left( \widehat{\overline{Z}}_{avg} \right)$

---

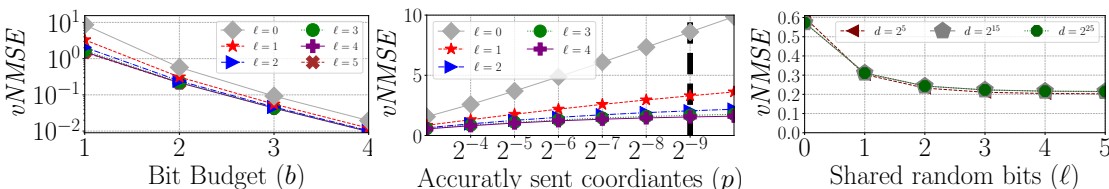

**Figure 1:** The *vNMSE* of QUIC-FL as a function of the bit budget, fraction $p$, and shared random bits $\ell$.

### 3.5 HADAMARD

Similarly to previous rotation-based compression algorithms Suresh et al. (2017); Vargaftik et al. (2021; 2022) we propose to use the Randomized Hadamard Transform (RHT) (Ailon & Chazelle, 2009) instead of uniform random rotations. Although RHT does not induce a uniform distribution on the sphere (and the coordinates are not exactly normally distributed), it is considerably more efficient to compute and, under mild assumptions, the resulting distribution is sufficiently close to the normal distribution Vargaftik et al. (2021). Here, we are interested in how using RHT affects the guarantees of our algorithm. We analyze how using RHT affects our guarantees, starting by noting that our algorithm remains unbiased *for any input vector*. However, adversarial inputs may (1) increase the probability that a rotated coordinate falls outside $[-T_p, T_p]$ and (2) increase the *vNMSE* as the coordinates' distribution deviates from the normal distribution. We show in Appendix D that QUIC-FL with RHT has similar guarantees as with random rotations, albeit somewhat weaker (constant factor increases in the fraction of accurately sent coordinates and *vNMSE*). We note that these guarantees are still stronger than those of DRIVE Vargaftik et al. (2021) and EDEN Vargaftik et al. (2022), which only prove RHT bounds for input vectors whose coordinates are sampled i.i.d. from a distribution with finite moments, and are not applicable to adversarial vectors. In practice, as shown in the evaluation, the actual performance is close to the theoretical results for uniform rotations; improving the bounds is left as future work. In our evaluation, we use QUIC-FL (Algorithm 2) with RHT-based vector rotation.

## 4 EVALUATION

### 4.1 THEORETICAL EVALUATION: *NMSE* AND SPEED MEASUREMENTS

**Parameter Selection.** We experiment with how the different parameters (number of quantiles $m$, the fraction of coordinates sent exactly $p$, the number of shared random bits $\ell$, etc.) affect the performance of our algorithm. As shown in Figure 1, introducing shared randomness decreases the *vNMSE* significantly compared with $\ell = 0$. Additionally, the benefit from adding each additional shared random bit diminishes, and the gain beyond $\ell = 4$ is negligible, especially for large $b$. Accordingly, we hereafter use $\ell = 6$ for $b = 1$, $\ell = 5$ for $b = 2$, and $\ell = 4$ for $b \in \{3, 4\}$. With respect to $p$, we determined $\frac{1}{512}$ as a good balance between the *vNMSE* and bandwidth overhead.

**Comparison to state of the art DME techniques.** Next, we compare the performance of QUIC-FL to the baseline algorithms in terms of *NMSE*, encoding speed, and decoding speed, using an NVIDIA 3080 RTX GPU machine with 32GB RAM and i7-10700K CPU @ 3.80GHz. Specifically, we compare with Hadamard Suresh et al. (2017), Kashin's representation Caldas et al. (2018); Safaryan et al. (2020), QSGD Alistarh et al. (2017), and EDEN Vargaftik et al. (2022). We evaluate two variants of

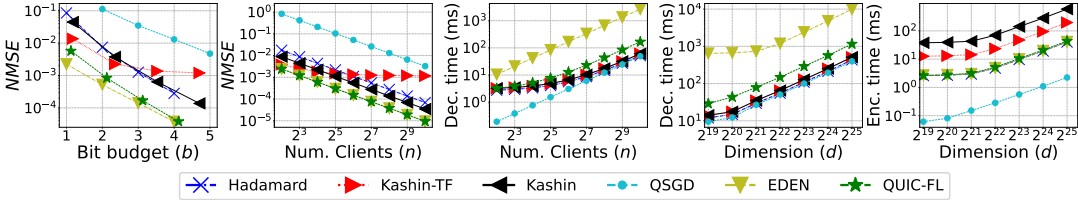

**Figure 2:** Comparison to alternatives with $n$ clients that have the same $LogNormal(0, 1)$ input vector Vargaftik et al. (2021; 2022). The default values are $n = 256$ clients, $b = 4$ bit budget, and $d = 2^{20}$ dimensions.

Kashin's representation: (1) The TensorFlow (TF) implementation Authors that, by default, limits the decomposition to three iterations, and (2) the theoretical algorithm that requires $O(\log(nd))$ iterations. As shown in Figure 2, QUIC-FL has the second-lowest $NMSE$, slightly higher than EDEN's, which has a far slower decode time. Further, QUIC-FL is significantly more accurate than approaches with similar speeds. We observed that the default TF configuration of Kashin's representation suffers from a bias, and therefore its $NMSE$ does not decrease inversely proportional to $n$. In contrast, the theoretical algorithm is unbiased but has a markedly higher encoding time. We observed similar trends for different $n$, $b$, and $d$ values. We consider the algorithms' bandwidth over all coordinates (e.g., with $b + \frac{64}{512}$ bits for QUIC-FL). Overall, the empirical measurements fall in line with the bounds in Table 1.

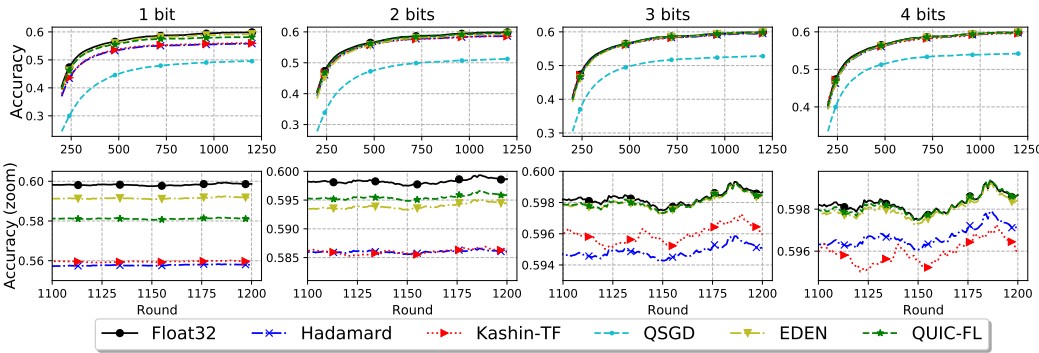

**Figure 3:** *FedAvg* over the Shakespeare next-word prediction task at various bit budgets (rows). We report training accuracy per round with a rolling mean window of 200 rounds. The second row zooms in on the last 100 rounds (QSGD is not included in the zoom since it performed poorly).

## 4.2 FEDERATED LEARNING EXPERIMENTS

**Next-word prediction.** We evaluate QUIC-FL over the Shakespeare next-word prediction task Shakespeare; McMahan et al. (2017) using an LSTM recurrent model. We run *FedAvg* McMahan et al. (2017) with the Adam server optimizer Kingma & Ba (2015) and sample $n = 10$ clients per round. We use the setup from the federated learning benchmark of Reddi et al. (2021), restated for convenience in Appendix E. Figure 3 shows how QUIC-FL compares with other compression schemes at various bit budgets. As shown, QUIC-FL is competitive with EDEN and nearly matches the accuracy of the uncompressed baseline for $b \geqslant 3$.

**Image classification.** We evaluate QUIC-FL against other schemes with 10 persistent clients over uniformly distributed CIFAR-10 and CIFAR-100 datasets Krizhevsky et al. (2009). We also evaluate *Count-Sketch* Charikar et al. (2002) (CS), often used for federated compression schemes (e.g., Ivkin et al. (2019)). For CIFAR-10 and CIFAR-100, we use ResNet-9 He et al. (2016) and ResNet-18 He et al. (2016), with learning rates of $0.1$ and $0.05$, respectively. For both datasets, the clients perform a single optimization step at each round. Our setting includes an SGD optimizer with a cross entropy loss criterion, a batch size of 128, and a bit budget $b = 1$.

The results are shown in Figure 4, with a rolling mean average window of 500 rounds. As shown, QUIC-FL is competitive with EDEN and the Float32 baseline and is more accurate than other methods.

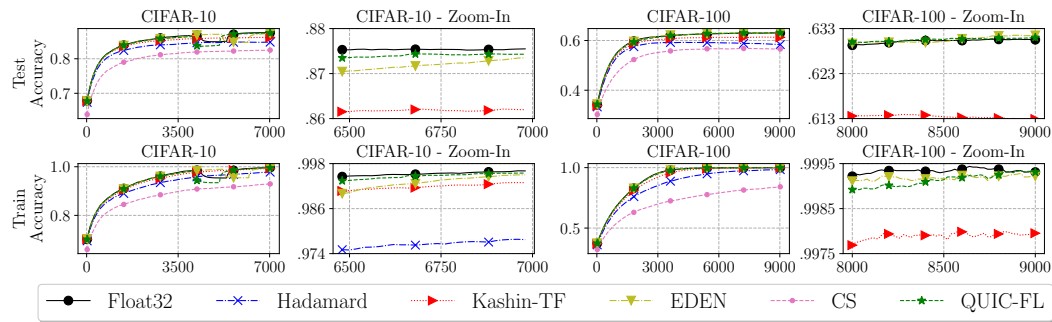

**Figure 4:** Train and test accuracy for CIFAR-10 and CIFAR-100 with 10 persistent clients (i.e., silos) and $b = 1$.

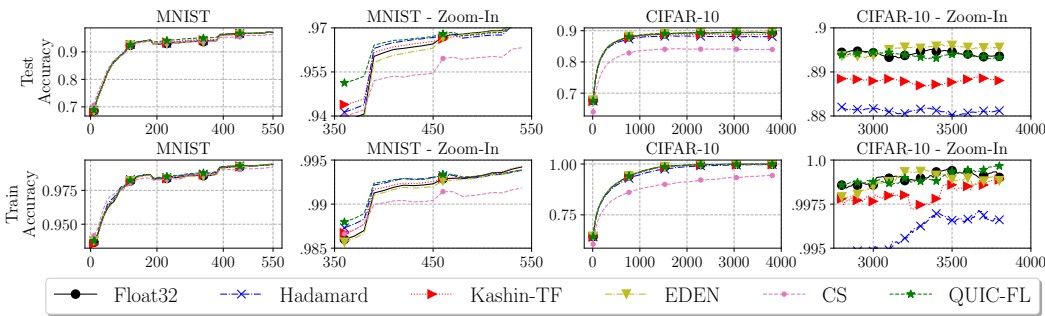

**Figure 5:** Cross-device federated learning of MNIST and CIFAR-10 with 50 clients ($b = 1$).

Next, we consider a highly heterogeneous cross-device setup with 50 clients over MNIST and CIFAR-10 datasets Krizhevsky et al. (2009); LeCun et al. (1998; 2010). For MNIST, each client stores only a single class of the dataset and trains LeNet-5 LeCun et al. (1998) with a learning rate of 0.05. For CIFAR-10, all clients have the same data distribution, and each trains ResNet-9 He et al. (2016) with a learning rate of 0.1. At each training round, 10 clients are randomly selected and perform training over 5 local steps. We use an SGD optimizer with a cross entropy loss criterion, a batch size of 128, and a bit budget $b = 1$.

Figure 5 shows the results with a rolling mean window of 200 rounds. Again, QUIC-FL is competitive with EDEN and the uncompressed baseline. Kashin-TF is less accurate followed by Hadamard.

**Additional evaluation** Due to lack of space, we defer additional evaluation results to Appendix F.

## 5 DISCUSSION

In this work, we presented QUIC-FL, a quick unbiased compression algorithm for federated learning. Both theoretically and empirically, QUIC-FL achieves an $NMSE$ that is comparable with the most accurate DME techniques, while allowing an asymptotically faster decode time.

We point out a few challenging directions for future work. QUIC-FL optimizes the worst-case error, and while it is compatible with orthogonal directions such as sparsification Konečný & Richtárik (2018); Vargaftik et al. (2022); Konečný et al. (2017); Fei et al. (2021), it is unclear how it would leverage potential correlations between coordinates Mitchell et al. (2022) or client vectors Davies et al. (2021). Another direction for future research is understanding how to incorporate non-linear aggregation functions, such as approximate geometric median, that have shown to improve the training robustness Pillutla et al. (2022).

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

## A  ALTERNATIVE COMPRESSION METHODS

This paper focused on the Distributed Mean Estimation (DME) problem where clients send lossily compressed gradients to a centralized server for averaging. While this problem is worthy of study on its own merits, we are particularly interested in applications to federated learning, where there are many variations and practical considerations, which have led to many alternative compression methods to be considered. For example, when the encoding and decoding time is less important, different approaches suggest to use entropy encodings such as Huffman or arithmetic encoding to improve the accuracy (e.g., Suresh et al. (2017); Vargaftik et al. (2022); Alistarh et al. (2017)). Intuitively, such encodings allow us to losslessly compress the lossily compressed vector to reduce its representation size, thereby allowing less aggressive quantization. However, we are unaware of available GPU-friendly entropy encoding implementation and thus such methods incur a significant time overhead.

Critically, for the basic DME problem, the assumption is that this is a one-shot process where the goal is to optimize the accuracy without relying on client-side memory. This model naturally fits cross-device federated learning, where different clients are sampled in each round. We focused on unbiased compression, which is standard in prior works Suresh et al. (2017); Konečný & Richtárik (2018); Vargaftik et al. (2021); Davies et al. (2021); Mitchell et al. (2022). However, if the compression error is low enough, and under some assumptions, SGD can be proven to converge even with biased compression Beznosikov et al. (2020).

In other settings, such as distributed learning or cross-silo federated learning, we may assume that clients are persistent and have a memory that keeps states between rounds. A prominent option to leverage such a state is to use Error Feedback (EF). In EF, clients can track the compression error and add it to the gradient computed in the consecutive round. This scheme is often shown to recover the model's convergence rate and resulting accuracy Seide et al. (2014); Alistarh et al. (2018); Richtárik et al. (2021) and enables biased compressors such as Top-$k$ Stich et al. (2018a).

An orthogonal proposal that works with persistent clients, which is also applicable with QUIC-FL, is to encode the difference between the current gradient and the previous one, instead of directly compressing the gradient Mishchenko et al. (2019); Gorbunov et al. (2021). Broadly speaking, this allows a compression error that is proportional to the size of the difference and not the gradient, and can decrease the error if consecutive gradients are similar to each other.

When running distributed learning in cluster settings, recent works show how in-network aggregation can accelerate the learning process Sapio et al. (2021); Lao et al. (2021); Segal et al. (2021). Intuitively, as switches are designed to move data at high speeds and recent advances in switch programmability enable it to easily perform simple aggregation operations like summation while processing the data.

Another line of work focuses on sparsifying the gradients before compressing them Fei et al. (2021); Stich et al. (2018b); Aji & Heafield (2017). Intuitively, in some learning settings, many of the coordinates are small. and we can improve the accuracy to bandwidth tradeoff by removing all small coordinates prior to compression. Another form of sparsification is random sampling, which allows us to avoid sending the coordinate indices Vargaftik et al. (2022); Konečný et al. (2017). We note that combining such approaches with QUIC-FL is straightforward, as we can compress the non-zero entries of the sparsified vectors.

Combining several techniques, including warm-up training, gradient clipping, momentum factor masking, momentum correction, and deep gradient compression Lin et al. (2018), reports savings of two orders of magnitude in the bandwidth required for distributed learning. Another promising orthogonal approach is to leverage shared randomness to get the clients' compression to yield errors in opposite directions, thus making them cancel out and lowering the overall NMSE Suresh et al. (2022).

The QUIC-FL algorithm, based on the output of the solver (see §3), uses *non-uniform* quantization, i.e., has quantization levels that are not uniformly spaced. Indeed, recent works observed that non-uniform quantization improves the estimation accuracy and accelerates the learning convergence Vargaftik et al. (2022); Ramezani-Kebrya et al. (2021).

We refer the reader to Kairouz et al. (2019); Konečný et al. (2017); Wang et al. (2021) for an extensive review of the current state of the art and challenges.

# B   QUIC-FL'S *vNMSE* PROOF

In this appendix, we analyze how the expected squared quantization error for a bounded-support normal random variables relates to the *vNMSE* of our algorithm. Namely, let $\chi = \mathbb{E}\left[\left(Z - \widehat{Z}\right)^2\right]$ denote the error of the quantization of a normal random variable $Z \sim \mathcal{N}(0, 1)$. Our analysis is general, covers the quantization methods presented in sections 3.2-3.3, but is applicable to any unbiased method that is used following a uniform random rotation. We show that QUIC-FL's *vNMSE* is essentially $\chi$ plus a small additional additive error term (arising because the rotation does not yield exactly normally distributed coordinates, as we have explained) that goes to 0 quickly as the dimension increases. We bound the additional error caused by using the Randomized Hadamard Transform in Section 3.5 and Appendix D.

**Theorem 1.** *It holds that:*

$$vNMSE \leqslant \chi + O\left(\sqrt{\frac{\log d}{d}}\right) \;.$$

*Proof.* The proof follows similar lines to that of Vargaftik et al. (2021; 2022). However, here the *vNMSE* expression is different and is somewhat simpler as it takes advantage of our *unbiased* quantization technique.

A rotation preserves a vector's euclidean norm. Thus, according to Algorithms 1 and 2 it holds that

$$\|x - \widehat{x}\|_2^2 = \|\mathcal{R}\left(x - \widehat{x}\right)\|_2^2 = \|\mathcal{R}\left(x\right) - \mathcal{R}\left(\widehat{x}\right)\|_2^2 = $$
$$\left\|\frac{\|x\|_2}{\sqrt{d}} \cdot \overline{Z} - \frac{\|x\|_2}{\sqrt{d}} \cdot \widehat{\overline{Z}}\right\|_2^2 = \frac{\|x\|_2^2}{d} \cdot \left\|\overline{Z} - \widehat{\overline{Z}}\right\|_2^2 \;. \tag{1}$$

Taking expectation and dividing by $\|x\|_2^2$ yields

$$vNMSE \triangleq \mathbb{E}\left[\frac{\|x - \widehat{x}\|_2^2}{\|x\|_2^2}\right] = \frac{1}{d} \cdot \mathbb{E}\left[\left\|\overline{Z} - \widehat{\overline{Z}}\right\|_2^2\right]$$
$$= \frac{1}{d} \cdot \mathbb{E}\left[\sum_{i=1}^{d}\left(\overline{Z}[i] - \widehat{\overline{Z}}[i]\right)^2\right] = \frac{1}{d} \cdot \sum_{i=1}^{d}\mathbb{E}\left[\left(\overline{Z}[i] - \widehat{\overline{Z}}[i]\right)^2\right] \;. \tag{2}$$

Let $\tilde{Z} = (\tilde{Z}_1, \ldots, \tilde{Z}_d)$ be a vector of independent $\mathcal{N}(0, 1)$ random variables. Then, the distribution of each coordinate $\overline{Z}[i]$ is given by $\tilde{Z}[i] \sim \sqrt{d} \cdot \frac{\tilde{Z}[i]}{\|\tilde{Z}\|_2}$ (e.g., see Vargaftik et al. (2021); Muller (1959)).

This means that all coordinates of $\overline{Z}$ and thus all coordinates of $\widehat{\overline{Z}}$ follow the same distribution. Thus, without loss of generality, we obtain

$$vNMSE \triangleq \mathbb{E}\left[\frac{\|x - \widehat{x}\|_2^2}{\|x\|_2^2}\right] = \mathbb{E}\left[\left(\overline{Z}[0] - \widehat{\overline{Z}}[0]\right)^2\right] = \mathbb{E}\left[\left(\frac{\sqrt{d}}{\|\tilde{Z}\|_2} \cdot \tilde{Z}[0] - \widehat{\overline{Z}}[0]\right)^2\right] \;. \tag{3}$$

For some $0 < \alpha < \frac{1}{2}$, denote the event

$$A = \left\{d \cdot (1 - \alpha) \leqslant \left\|\tilde{Z}\right\|_2^2 \leqslant d \cdot (1 + \alpha)\right\} \;.$$

Let $A^c$ be the complementary event of $A$. By Lemma D.2 in Vargaftik et al. (2022) it holds that $\mathbb{P}(A^c) \leqslant 2 \cdot e^{-\frac{\alpha^2}{8} \cdot d}$. Also, by the law of total expectation

$$\mathbb{E}\left[\left(\frac{\sqrt{d}}{\left\|\tilde{Z}\right\|_2}\cdot\tilde{Z}[0]-\widehat{\tilde{Z}}[0]\right)^2\right]\leqslant$$

$$\mathbb{E}\left[\left(\frac{\sqrt{d}}{\left\|\tilde{Z}\right\|_2}\cdot\tilde{Z}[0]-\widehat{\tilde{Z}}[0]\right)^2\bigg|A\right]\cdot\mathbb{P}(A)+\mathbb{E}\left[\left(\frac{\sqrt{d}}{\left\|\tilde{Z}\right\|_2}\cdot\tilde{Z}[0]-\widehat{\tilde{Z}}[0]\right)^2\bigg|A^c\right]\cdot\mathbb{P}(A^c)\leqslant \quad (4)$$

$$\mathbb{E}\left[\left(\frac{\sqrt{d}}{\left\|\tilde{Z}\right\|_2}\cdot\tilde{Z}[0]-\widehat{\tilde{Z}}[0]\right)^2\bigg|A\right]\cdot\mathbb{P}(A)+M\cdot\mathbb{P}(A^c)\,,$$

where $M=\left(vNMSE_{max}\right)^2$ and $vNMSE_{max}$ is the maximal value at the server's reconstruction table (i.e., $\max(r)$) which is a constant that is *independent* of the vector's dimension. Next,

$$\mathbb{E}\left[\left(\frac{\sqrt{d}}{\left\|\tilde{Z}\right\|_2}\cdot\tilde{Z}[0]-\widehat{\tilde{Z}}[0]\right)^2\bigg|A\right]=\mathbb{E}\left[\left(\left(\tilde{Z}[0]-\widehat{\tilde{Z}}[0]\right)+\left(\frac{\sqrt{d}}{\left\|\tilde{Z}\right\|_2}-1\right)\cdot\tilde{Z}[0]\right)^2\bigg|A\right]=$$

$$\mathbb{E}\left[\left(\tilde{Z}[0]-\widehat{\tilde{Z}}[0]\right)^2\bigg|A\right]+2\cdot\mathbb{E}\left[\left(\tilde{Z}[0]-\widehat{\tilde{Z}}[0]\right)\cdot\left(\frac{\sqrt{d}}{\left\|\tilde{Z}\right\|_2}-1\right)\cdot\tilde{Z}[0]\bigg|A\right]+$$

$$\mathbb{E}\left[\left(\left(\frac{\sqrt{d}}{\left\|\tilde{Z}\right\|_2}-1\right)\cdot\tilde{Z}[0]\right)^2\bigg|A\right]$$

$$(5)$$

Also,

$$\mathbb{E}\left[\left(\tilde{Z}[0]-\widehat{\tilde{Z}}[0]\right)\cdot\left(\frac{\sqrt{d}}{\left\|\tilde{Z}\right\|_2}-1\right)\cdot\tilde{Z}[0]\bigg|A\right]\cdot\mathbb{P}(A)\leqslant$$

$$\left(\frac{1}{\sqrt{1-\alpha}}-1\right)\cdot\left|\mathbb{E}\left[\left(\tilde{Z}[0]-\widehat{\tilde{Z}}[0]\right)\cdot\tilde{Z}[0]|A\right]\cdot\mathbb{P}(A)\right|\leqslant \quad (6)$$

$$\left(\frac{1}{\sqrt{1-\alpha}}-1\right)\cdot\left|\mathbb{E}\left[\left(\tilde{Z}[0]\right)^2-\widehat{\tilde{Z}}[0]\cdot\tilde{Z}[0]\bigg|A\right]\cdot\mathbb{P}(A)\right|\leqslant$$

$$\left(\frac{1}{\sqrt{1-\alpha}}-1\right)\cdot1+\left(\frac{1}{\sqrt{1-\alpha}}-1\right)\cdot\frac{1}{\sqrt{1-\alpha}}=\frac{\alpha}{1-\alpha}\leqslant2\alpha\,.$$

Here, we used that $\mathbb{E}\left[\left(\tilde{Z}[0]\right)^2|A\right]\cdot\mathbb{P}(A)\leqslant\mathbb{E}\left[\left(\tilde{Z}[0]\right)^2\right]=1$ and that $\mathbb{E}\left[\widehat{\tilde{Z}}[0]\cdot\tilde{Z}[0]|A\right]\cdot\mathbb{P}(A)=$
$\mathbb{E}\left[\mathbb{E}\left[\widehat{\tilde{Z}}[0]\cdot\tilde{Z}[0]|A,\tilde{Z}\right]\right]\cdot\mathbb{P}(A)=\mathbb{E}\left[\frac{\sqrt{d}}{\left\|\tilde{Z}\right\|_2}\cdot\left(\tilde{Z}[0]\right)^2\bigg|A\right]\cdot\mathbb{P}(A)\leqslant\frac{1}{\sqrt{1-\alpha}}\cdot\mathbb{E}\left[\left(\tilde{Z}[0]\right)^2\right]=\frac{1}{\sqrt{1-\alpha}}.$
Next, we similarly obtain

$$\mathbb{E}\left[\left(\left(\frac{\sqrt{d}}{\left\|\tilde{Z}\right\|_2}-1\right)\cdot\tilde{Z}[0]\right)^2\bigg|A\right]\cdot\mathbb{P}(A)\leqslant\left(\frac{1}{\sqrt{1-\alpha}}-1\right)+\left(1-\frac{1}{\sqrt{1+\alpha}}\right)\leqslant2\alpha. \quad (7)$$

Thus,

$$vNMSE\leqslant\mathbb{E}\left[\left(\tilde{Z}[0]-\widehat{\tilde{Z}}[0]\right)^2\right]+4\alpha+2\cdot e^{-\frac{\alpha^2}{8}\cdot d}\cdot M\,. \quad (8)$$

Setting $\alpha=\sqrt{\frac{8\log d}{d}}$ yields $vNMSE\leqslant\mathbb{E}\left[\left(\tilde{Z}[0]-\widehat{\tilde{Z}}[0]\right)^2\right]+O\left(\sqrt{\frac{\log d}{d}}\right).$

Finally, since $\widetilde{Z}[0] \sim \mathcal{N}(0,1)$, we can write

$$vNMSE \leqslant \mathbb{E}\left[\left(Z - \widehat{Z}\right)^2\right] + O\left(\sqrt{\frac{\log d}{d}}\right). \quad \square$$

# C INTERPOLATING THE SOLVER'S SOLUTION

## C.1 EXAMPLES, INTUITION AND PSEUDOCODE

Based on our examination of solver outputs, we determined an alternative approach that does not stochastically quantize each coordinate to a quantile as above and empirically performs better.

We explain the process first considering an example. We consider the setting of $p = \frac{1}{512}$ ($T_p \approx 3.097$), $m = 512$ quantiles, $b = 2$ bits per coordinate, and $\ell = 2$ bits of shared randomness. One optimal solution for the server is given below:[6]

|         | $x = 0$ | $x = 1$ | $x = 2$ | $x = 3$ |
| ------- | ------- | ------- | ------- | ------- |
| $h = 0$ | -5.48   | -1.23   | **0.164**  | 1.68    |
| $h = 1$ | -3.04   | -0.831  | **0.490**  | 2.18    |
| $h = 2$ | -2.18   | **-0.490** | 0.831   | 3.04    |
| $h = 3$ | -1.68   | **-0.164** | 1.23    | 5.48    |

**Table 2:** Optimal server values $(r_{h,x})$ for $x \in \mathcal{X}_2, H \in \mathcal{H}_2$ when $p = 1/512$ and $m = 512$, rounded to 3 significant digits. For example, when $Z = 0$, the server will estimate one of the values in bold based on the shared randomness and the message received from the client.

Given this table, by symmetry, if $Z = 0$ we can send $X = \begin{cases} 1 & \text{If } H \leqslant 1 \\ 2 & \text{Otherwise} \end{cases}$, which is formally

written as $S_x(H, 0) = \begin{cases} 1 & \text{If } (x = 1 \wedge H \leqslant 1) \vee (x = 2 \wedge H > 1) \\ 0 & \text{Otherwise} \end{cases}$. Indeed, we have that $\mathbb{E}\left[\widehat{Z}\right] =$

$\frac{1}{4} \sum_h r_{h,X} = 0$. Now, suppose that $Z > 0$ (the negative case is symmetric); the client can increase the server estimate's expected value (compared with the above choice of $X$) by moving probability mass to larger $x$ values for some (or all) of the options for $H$. For any $Z \in (-T_p, T_p)$, there are infinitely many client alternatives that would yield an unbiased estimate. For example, if $Z = 0.1$, below are two client options (rounded to one significant digit):

$$S'_x(H, 0.1) \approx \begin{cases} 1 & \text{If } (x = 1 \wedge H \leqslant 2) \\ 0.6 & \text{If } (x = 2 \wedge H = 3) \\ 0.4 & \text{If } (x = 3 \wedge H = 3) \\ 0 & \text{Otherwise} \end{cases}, \quad S''_x(H, 0.1) \approx \begin{cases} 1 & \text{If } (x = 2 \wedge H \leqslant 1) \vee (x = 1 \wedge H = 3) \\ 0.3 & \text{If } (x = 1 \wedge H = 2) \\ 0.7 & \text{If } (x = 2 \wedge H = 2) \\ 0 & \text{Otherwise} \end{cases}$$

Note that while both $S'$ and $S''$ produce unbiased estimates, their expected squared errors differ. Further, since $0.1 \notin \mathcal{Q}_m$, the solver's output does not directly indicate what is the optimal client's algorithm, even if the server table is fixed. Unlike Algorithm 1, which stochastically quantizes $Z$ to either $q^-$ or $q^+$, we studied the solver's output $\{s_{h,q,x}\}_{h,q,x}$ to interpolate the client to non-quantile values.

The approach we take corresponds to the following process. We move probability mass from the *leftmost, then uppermost* entry with mass to its right neighbor in the server table. So, for example, in Table 2, as $Z$ increases from 0 we first move mass from the entry $x = 1, h = 2$ to the entry $x = 2, h = 2$. That is, the client, based on its private randomness, increases the probability of message $x = 2$ and decreases the probability of message $x = 1$ when $h = 2$. The amount of mass moved is always chosen to maintain unbiasedness. At some point, as $Z$ increases, all of the probability mass will have moved, and then we start moving mass from $x = 1, h = 3$ similarly.

---

[6] A crucial ingredient in getting a human-readable solution from the solver is that we, without loss of generality, force monotonicity in both $h$ and $x$, i.e., $(x \geqslant x') \wedge (h \geqslant h') \implies r_{h,x} \geqslant r_{h',x'}$. Further, note that Table 2 is symmetric. We found tables were symmetric for small $\ell, m$, and then forced symmetry in order to reduce model size for larger values. We use this symmetry in our interpolation.

---

**Algorithm 2** QUIC-FL

---

**Client $c$:**
1: Compute $\overline{Z}_c = \frac{\sqrt{d}}{\|x_c\|_2} \cdot \mathcal{R}(x_c)$.
2: Compute $S(H_c, \overline{Z}_c)$ as in equation 9 given in Appendix C.2
3: Sample $\overline{X}_c \sim S(H_c, \overline{Z}_c)$
4: Send $\left(\|x_c\|_2, X_c\right)$ to server

**Server:**
1: $\forall c$ : Compute $\widehat{\overline{Z}}_c = r_{\overline{H}_c, \overline{X}_c}$
2: Compute $\widehat{\overline{Z}}_{avg} = \frac{1}{n} \cdot \frac{1}{\sqrt{d}} \cdot \sum_{c=1}^n \|x_c\|_2 \cdot \widehat{\overline{Z}}_c$
3: Estimate $\hat{x}_{avg} = \mathcal{R}^{-1}\left(\widehat{\overline{Z}}_{avg}\right)$

---

(And subsequently, from $x = 2, h = 0$ and so on.) This process is visualized in Figure 6. Note that $S_x(h, z)$ values are piecewise linear as a function of $z$, and further, these values either go from 0 to 1, 1 to 0, or 0 to 1 and back again (all of which follow from our description). We can turn this description into formulae, and we defer this mathematical interpretation to Appendix C. The final algorithm, named QUIC-FL, is given by Algorithm 2 (based on the formula given in the appendix).

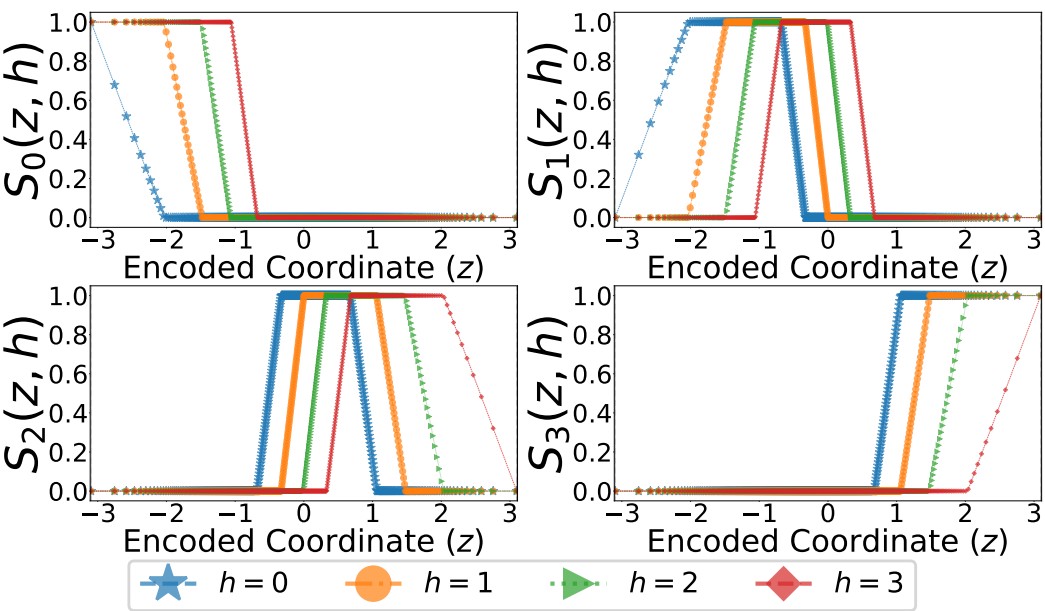

**Figure 6:** The solver's client algorithm (for $b = \ell = 2, m = 512, p = \frac{1}{512}$) for the quantiles $\{s_{h,q,x}\}_{h,q,x}$. Markers correspond to quantiles in $\mathcal{Q}_m$, and the lines illustrate our interpolation.

## C.2 DERIVATION OF EQUATIONS FOR ALGORITHM 2

As we described in §C.1 through an example, and as illustrated by Figure 6, we have found through examining the solver's solutions, for our parameter range, that the optimal approach for the client has a structure that we can generalize. This allows us to readily interpolate the algorithm to non-quantile values $Z \notin \mathcal{Q}_m$ without stochastically quantizing the coordinate to a quantile.

Recall the example from §C.1 with $\ell = b = 2$. There, we had the server table:

|       | $x = 0$ | $x = 1$ | $x = 2$ | $x = 3$ |
|-------|---------|---------|---------|---------|
| $h = 0$ | -5.48 | -1.23 | **0.164** | 1.68 |
| $h = 1$ | -3.04 | -0.831 | **0.490** | 2.18 |
| $h = 2$ | -2.18 | **-0.490** | 0.831 | 3.04 |
| $h = 3$ | -1.68 | **-0.164** | 1.23 | 5.48 |

When $Z = 0$, the client would send $X = 2$ if $H \leqslant 1$ and $X = 1$ otherwise, leading to $\mathbb{E}\left[\widehat{Z}\right] = 0$. Now consider $Z = 0.1$; the client can increase the expected server's estimate by changing its behaviour for the leftmost and then uppermost cell, namely $X = 1, H = 2$. Specifically, by following

$$
S''_x(H, 0.1) \approx \begin{cases} 1 & \text{If } (x = 2 \wedge H \leqslant 1) \vee (x = 1 \wedge H = 3) \\ 0.3 & \text{If } (x = 1 \wedge H = 2) \\ 0.7 & \text{If } (x = 2 \wedge H = 2) \\ 0 & \text{Otherwise} \end{cases},
$$

In general, by applying the monotonicity constraints[6] we observed a common pattern in the optimal solution found by the solver for any $b$ and $\ell$ (in the range we tested). Namely, when the server table is monotone, the optimal solution *deterministically* selects the message to send in all but (at most) one shared randomness value. For example, $S''_x$ above deterministically selects the message if $H \neq 2$ (sending 1 if $H = 3$ and 2 if $H \in \{0, 1\}$) and stochastically selects between $x = 1$ and $x = 2$ when $H = 2$. Furthermore, the shared randomness value in which we should stochastically select the message is easy to calculate. Specifically, let $\underline{x}(Z) \in \mathcal{X}_b$ denote the maximal value such that sending $\underline{x}(Z)$ for all $H$ would result in not overestimating $Z$ in expectation (that is: $\underline{x}(Z) = \max\left\{x \in \mathcal{X}_b \mid \frac{1}{2^\ell} \cdot \sum_{h \in \mathcal{H}_\ell} r_{h,x} \leqslant Z\right\}$). We note that there must exists such a $\underline{x}(Z)$ because, by design, the solver's output for the optimal solution would always satisfy that $\frac{1}{2^\ell} \cdot \sum_{h=0}^{2^\ell} r_{h,0} = -T$ and $\frac{1}{2^\ell} \cdot \sum_{h=0}^{2^\ell} r_{h,2^b-1} = T$. (Otherwise, the solution would either be infeasible or suboptimal.) In particular, this means that we get $\underline{x}(Z) \in \mathcal{X}_b \backslash \left\{2^b - 1\right\}$ for all $Z \in [-T, T)$.

Next, let $\underline{h}(Z) \in \mathcal{H}_\ell$ denote the maximal value for which sending $\underline{x}(Z) + 1$ for all $H < \underline{h}$ and $\underline{x}(Z)$ for $H \geqslant \underline{h}$ would underestimate $Z$ in expectation (for convenience, we consider $r_{h,2^b} = \infty$, for all $h \in \mathcal{H}_\ell$). Formally:

$$
\underline{h}(Z) = \max\left\{ h \in \mathcal{H} \;\middle|\; \frac{1}{2^\ell} \cdot \left( \sum_{h'=0}^{h-1} r_{h',\underline{x}(Z)+1} + \sum_{h'=h}^{2^\ell-1} r_{h',\underline{x}(Z)} \right) \leqslant Z \right\} .
$$

For example, consider $Z = 0.1$ in the $b = \ell = 2$ example above. We have that $\underline{x}(0.1) = 1$ since $\frac{1}{2^2} \cdot \sum_{h \in \mathcal{H}_2} r_{h,1} \leqslant 0.1$ and $\frac{1}{2^2} \cdot \sum_{h \in \mathcal{H}_2} r_{h,2} > 0.1$. We also have that $\underline{h}(Z) = 2$ as $\frac{1}{4} \cdot (0.164 + 0.49 + (-0.49) + (-0.164)) \leqslant 0.1$ and $\frac{1}{4} \cdot (0.164 + 0.49 + 0.831 + (-0.164)) > 0.1$. Similarly, for $Z = 3$ we get $\underline{x}(3) = 2$ and $\underline{h}(3) = 3$.

The interpolated algorithm works as follows: If $H < \underline{h}$, it deterministically sends $\underline{x}(Z) + 1$, i.e., $S_x(H, Z) = \begin{cases} 1 & \text{If } (x = \underline{x}(Z) + 1) \\ 0 & \text{otherwise} \end{cases}$. If $H > \underline{h}$, it sends $\underline{x}(Z)$, i.e., $S_x(H, Z) = \begin{cases} 1 & \text{If } (x = \underline{x}(Z)) \\ 0 & \text{otherwise} \end{cases}$.

Finally, for $H = \underline{h}$, it stochastically selects between $\underline{x}(Z)$ and $\underline{x}(Z) + 1$. However, notice that the expected value of this quantization (given that $H = \underline{h}$) may *not* be exactly $Z$. Namely, as the algorithm needs to satisfy $\mathbb{E}[\widehat{Z} \mid Z] = \frac{1}{2^\ell} \sum_{h \in \mathcal{H}_\ell} \mathbb{E}[\widehat{Z} \mid H = h] = Z$, we require that

$$
\mu \triangleq \mathbb{E}[\widehat{Z} \mid H = \underline{h}] = 2^\ell \cdot Z - \sum_{h=0}^{\underline{h}-1} r_{h,\underline{x}(Z)+1} + \sum_{h=\underline{h}+1}^{2^\ell-1} r_{h,\underline{x}(Z)} .
$$

Knowing the desired expectation, the overall client's algorithm is then defined as:

$$
S_x(H, Z) = \begin{cases} 1 & \text{If } (x = \underline{x}(Z) \wedge H > \underline{h}) \vee (x = \underline{x}(Z) + 1 \wedge H < \underline{h}) \\ \frac{\mu - r_{h,\underline{x}(Z)}}{r_{h,\underline{x}(Z)+1} - r_{h,\underline{x}(Z)}} & \text{If } (x = \underline{x}(Z) + 1 \wedge H = \underline{h}) \\ \frac{r_{h,\underline{x}(Z)+1} - \mu}{r_{h,\underline{x}(Z)+1} - r_{h,\underline{x}(Z)}} & \text{If } (x = \underline{x}(Z) \wedge H = \underline{h}) \\ 0 & \text{Otherwise} \end{cases}. \quad (9)
$$

Indeed, by our choice of $\mu$, the algorithm is guaranteed to be unbiased for all $Z \in [-T, T]$. This gives the final algorithm, whose pseudo-code appears in Algorithm 2. As in Algorithm 1, each client $c$ rotates its vector and scales it by $\frac{\sqrt{d}}{\|x_c\|_2}$ to get a vectors whose coordinates' distribution approach i.i.d. $\mathcal{N}(0, 1)$. Next, for each entry $\overline{Z}_c[i]$ the client computes $\underline{x}(\overline{Z}_c[i]), \underline{h}(\overline{Z}_c[i])$, and then it computes $S(H_c, \overline{Z}_c[i])$ as shown in Eq equation 9. Finally, it samples a message for each coordinate $\overline{X}_c \sim S(H_c, \overline{Z}_c[i])$ that it sends to the server together with the norm $\|x_c\|_2$.

# D  PERFORMANCE OF QUIC-FL WITH THE RANDOMIZED HADAMARD TRANSFORM

As described earlier, while ideally we would like to use a fully random rotation on the $d$-dimensional sphere as the first step to our algorithms, this is computationally expensive. Instead, we suggest using a randomized Hadamard transform (RHT), which is computationally more efficient. We formally show below that we maintain some performance bounds using a single RHT.

We note that some works suggest using two or three successive randomized Hadamard transforms to obtain something that should be closer to a uniform random rotation Yu et al. (2016); Andoni et al. (2015). This naturally takes more computation time. In our case, and in line with previous works Vargaftik et al. (2021; 2022), we find empirically that one RHT appears to suffice. Unlike these works, our algorithm remains provably unbiased and maintains strong $NMSE$ guarantees in this case. Determining better provable bounds using two or more RHTs is left as an open problem.

**Theorem 2.** *Let $x \in \mathbb{R}^d$, let $\mathcal{R}_{RHT}(x)$ be its randomized Hadamard transform, and let $\mathfrak{Z} = \frac{\sqrt{d}}{\|x\|_2} \mathcal{R}_{RHT}(x)[i]$ be a coordinate in the transformed vector. For any $p$, $\Pr\left[\mathfrak{Z} \notin [-T_p, T_p]\right] \leqslant 3.2p$.*

*Proof.* Follows from the following theorem.

**Theorem 3** (Bentkus & Dzindzalieta (2015)). *Let $\epsilon_1, \ldots, \epsilon_d$ be i.i.d. Radamacher random variables and let $a \in \mathbb{R}^d$ such that $\|a\|_2^2 \leqslant 1$. For any $t \in \mathbb{R}$, $\Pr\left[\sum_{i=1}^{d} a_i \cdot \epsilon_i \geqslant t\right] \leqslant \frac{\Pr[Z \geqslant t]}{4 \Pr[Z \geqslant \sqrt{2}]} \approx 3.1787 \Pr[Z \geqslant t]$, for $Z \sim \mathcal{N}(0, 1)$.* $\square$

**Theorem 4.** *Fix $p = \frac{1}{512}$; let $x \in \mathbb{R}^d$, let $\mathcal{R}_{RHT}(x)$ be its randomized Hadamard transform, and let $\mathfrak{Z} = \frac{\sqrt{d}}{\|x\|_2} \mathcal{R}_{RHT}(x)[i]$ be a coordinate in the transformed vector. Denoting by $E_b = \mathbb{E}\left[(\mathfrak{Z} - \widehat{\mathfrak{Z}_b})^2\right]$ the mean squared error using $b$ bits per coordinate, we have $E_1 \leqslant 4.831$, $E_2 \leqslant 0.692$, $E_3 \leqslant 0.131$, $E_4 \leqslant 0.0272$.*

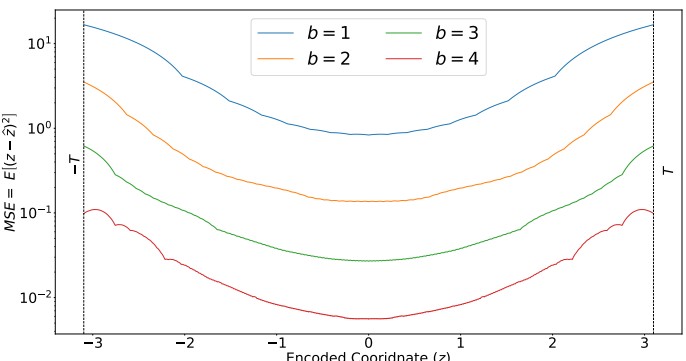

**Figure 7:** Expected squared error as a function of the value of $\mathfrak{Z}$ (for $p = \frac{1}{512}, m = 512$).

*Proof.* We present an approach to bound the MSE of encoding the coordinate $\mathfrak{Z}$, leveraging Theorem 3. Since we believe that it provides only a loose bound, we do not optimize the argument beyond showing the technique. Since the MSE, as a function of $\mathfrak{Z}$, is symmetric around $0$ (as illustrated in Figure 7), we analyze the $\mathfrak{Z} \geqslant 0$ case.

The first option is to split $[0, T]$ into intervals, e.g., $I_1 = [0, 1.5]$, $I_2 = (1.5, 2.2]$, $I_3 = (2.2, T)$. Using Theorem 3, we get that $P_1 \triangleq \Pr[\mathfrak{Z} \notin [0, 1.5]] \leqslant 3.2 \Pr[Z \notin [0, 1.5]] \leqslant 0.427$ and similarly, $P_2 \triangleq \Pr[\mathfrak{Z} \notin [0, 2.2]] \leqslant 3.2 \Pr[Z \notin [0, 2.2]] \leqslant 0.089$. Next, we provide the maximal error for each bit budget $b$ and such interval:

|              | $b = 1$ | $b = 2$ | $b = 3$ | $b = 4$ |
|--------------|---------|---------|---------|---------|
| $[0, 1.5]$   | 2.063   | 0.267   | 0.056   | 0.0134  |
| $[1.5, 2.2]$ | 6.39    | 0.67    | 0.128   | 0.0285  |
| $[2.2, T]$   | 16.73   | 3.51    | 0.617   | 0.11    |

**Table 3:** For each interval $I$ and bit budget $b$, the maximal MSE, i.e., $\max_{z \in I} \mathbb{E}\left[(z - \hat{z})^2\right]$.

Note that for any $b \in \{1, 2, 3, 4\}$, the MSEs in $I_3$ are strictly larger than those in $I_2$ which are strictly larger than those in $I_1$. This allows us to derive formal bounds on the error. For example, for $b = 1$, we have that the error is bounded by

$$E_1 \leqslant (1 - 0.427) \cdot 2.063 + (0.427 - 0.089) \cdot 6.39 + 0.089 \cdot 16.73 \leqslant 4.831.$$

Repeating this argument, we also obtain:

$$E_2 \leqslant (1 - 0.427) \cdot 0.267 + (0.427 - 0.089) \cdot 0.67 + 0.089 \cdot 3.51 \leqslant 0.692$$
$$E_3 \leqslant (1 - 0.427) \cdot 0.056 + (0.427 - 0.089) \cdot 0.128 + 0.089 \cdot 0.617 \leqslant 0.131$$
$$E_4 \leqslant (1 - 0.427) \cdot 0.0134 + (0.427 - 0.089) \cdot 0.0285 + 0.089 \cdot 0.11 \leqslant 0.0272. \qquad \square$$

## E  SHAKESPEARE EXPERIMENTS DETAILS

The Shakespeare next-word prediction task was first suggested in McMahan et al. (2017) to naturally simulate a realistic heterogeneous federated learning setting. Its dataset consists of 18,424 lines of text from Shakespeare plays Shakespeare partitioned among the respective 715 speakers (i.e., clients). We train a standard LSTM recurrent model Hochreiter & Schmidhuber (1997) with $\approx 820K$ parameters and follow precisely the setup described in Reddi et al. (2021) for the Adam server optimizer case. We restate the hyperparameters for convenience in Table 4.

| Task        | Clients per round | Rounds | Batch size | Client lr | Server lr | Adam's $\epsilon$ |
|-------------|-------------------|--------|------------|-----------|-----------|-------------------|
| Shakespeare | 10                | 1200   | 4          | 1         | $10^{-2}$ | $10^{-3}$         |

**Table 4:** Hyperparameters for the Shakespeare next-word prediction experiments.

## F  ADDITIONAL EVALUATION

Our code appears in the supplementary material and will be released as open source upon publication.

### F.1  DISTRIBUTED POWER ITERATION

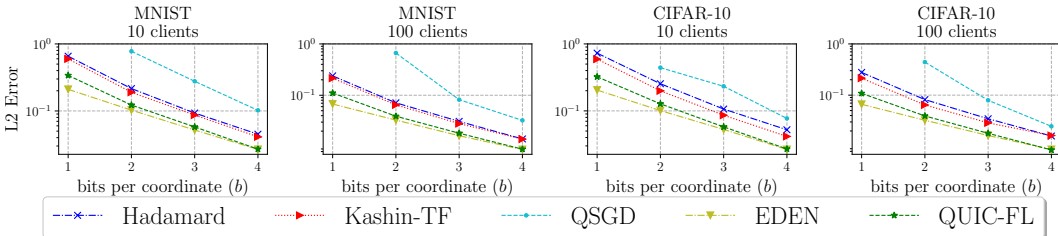

**Figure 8:** Distributed power iteration of MNIST and CIFAR-10 with 10 and 100 clients.

We simulate 10 clients that distributively compute the top eigenvector in a matrix (i.e., the matrix rows are distributed among the clients). Particularly, each client executes a power iteration, compresses its top eigenvector, and sends it to the server. The server updates the next estimated eigenvector by the averaged diffs (of each client to the eigenvector from the previous round) and scales it by a learning rate of 0.1. Then, the estimated eigenvector is sent by the server to the clients and the next round can begin.

Figure 8 presents the L2 error of the obtained eigenvector by each compression scheme when compared to the eigenvector that is achieved without compression. The results cover bit budget $b$ from one bit to four bits for both MNIST and CIFAR-10 Krizhevsky et al. (2009); LeCun et al. (1998; 2010) datasets. Each distributed power iteration simulation is executed for 50 rounds for the MNIST dataset and for 200 rounds for the CIFAR-10 dataset.

As shown, QUIC-FL has an accuracy that is competitive with that of EDEN (especially for $b \geqslant 2$), and considerably better than other algorithms that offer fast decoding time. Also, Kashin-TF is not unbiased (as illustrated by Figure 1), and is therefore less competitive for a larger number of clients.

### F.2 FEDERATED LEARNING: EXTENDED COMPARISON WITH QSGD

We repeat the experiments from Figure 4 and Figure 5, adding another curve for QSGD, which uses twice the bandwidth of the other algorithms (one bit for sign, and another for the stochastic quantization). As shown in figures 9 and 10, even with the additional bandwidth, QSGD's accuracy is lower than all algorithms except CS. We note that QSGD also has a more accurate variant that uses variable length encoding Alistarh et al. (2017). However, it is not GPU-friendly, and therefore, as with other variable length encoding schemes as we have discussed previously, we do not include it in the experiment.

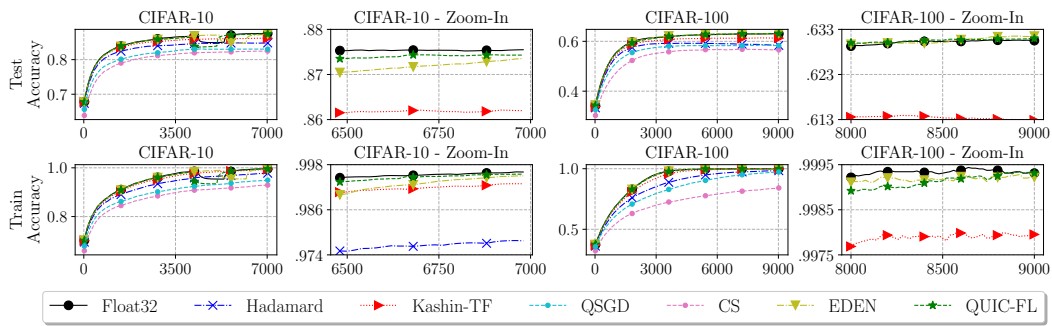

**Figure 9:** Train and test accuracy for CIFAR-10 and CIFAR-100 with 10 persistent clients (i.e., silos) and $b = 1$.

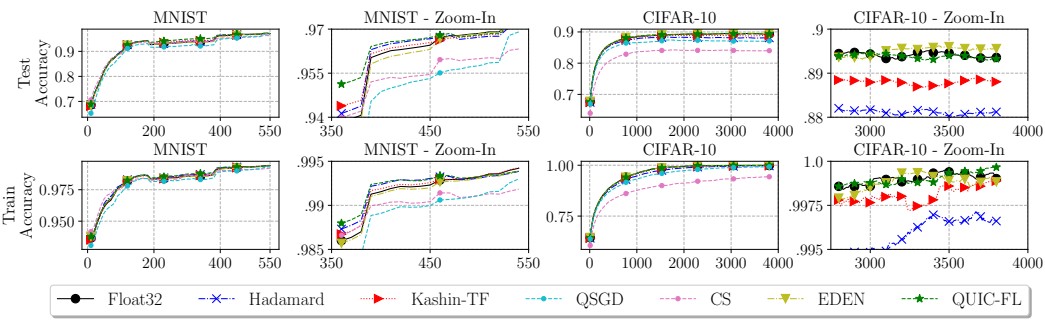

**Figure 10:** Cross-device federated learning of MNIST and CIFAR-10 with 50 clients and $b = 1$.

## G ANALYSIS OF THE BOUNDED STOCHASTIC QUANTIZATION TECHNIQUE

In this appendix, we analyze the Bounded Stochastic Quantization (BSQ) approach that sends all coordinates outside a range $[-T, T]$ exactly and performs a standard stochastic quantization for the rest.

Let $p \in (0, 1)$ and denote $T_p = \frac{\|x\|_2}{\sqrt{d \cdot p}}$; notice that there can be at most $d \cdot p$ coordinates outside $[-T_p, T_p]$. Using $b$ bits, we split this range into $2^b - 1$ intervals of size $\frac{2T_p}{2^b - 1}$, meaning that each coordinate's expected squared error is at most $\left(\frac{2T_p}{2^b - 1}\right)^2 / 4$. The MSE of the algorithm is therefore

bounded by

$$\mathbb{E}\left[\|x - \hat{x}\|_2^2\right] = d \cdot \left(\frac{2T_p}{2^b - 1}\right)^2 /4 = \frac{\|x\|_2^2}{p \cdot (2^b - 1)^2}.$$

This gives a result of

$$vNMSE \leqslant \frac{1}{p \cdot (2^b - 1)^2}.$$

Let $r$ be the representation length of each coordinate in the input vector (e.g., $r = 32$ for single-precision floats, $r = 16$ for half-precision, or $r = 64$ for double precision), we get that BSQ sends a message with less than $p \cdot r + b$ bits per coordinate. Further, this method has $O(d)$ time for encoding and decoding and is GPU-friendly.

