# OpenReview forum: "QUIC-FL: : Quick Unbiased Compression for Federated Learning"
_ICLR.cc/2023/Conference — Submitted to ICLR 2023_

### Official Review · Reviewer_V1XF · 2022-10-24

**Confidence:** 4
**Correctness:** 2
**Technical Novelty And Significance:** 2
**Empirical Novelty And Significance:** 2
**Recommendation:** 3

**Clarity, Quality, Novelty And Reproducibility:**

The paper is relatively well-written. The techniques are mostly based on Vargaftik et al. (2021; 2022).  To improve reproducibility, the authors should elaborate on the hyperparameters used for baselines in Section 4.

**Strength And Weaknesses:**

Strength: The paper is relatively well-written. It also interesting that the authors provide simple examples such as those in Sections 3.3 and 3.4 to help the readers understand the method.

Weaknesses: Unfortunately, I think the main weakness is that there are major technical issues in the paper, which will be discussed in the following:


------

1- "transmitting a “bounded-support” normal random variable $Z$, $Z\in[-T_p, T_p]$ using $b$ bits"

Given the bounded support, the distrubution will not be normal anymore. It will be "truncated normal" with different probability distribution function and moments. I am not sure about the correctness of the analyses in this paper since the authors assume normal distribution for coordinates after the rotation.

For example, the expected squared error in page 4 and optimal unbiased quantization problem in page 5 are not correct because the distribution of $Z$ is truncated normal distribution.

------

2- Regarding, the dimension-indpendent upper bounds on the normalized mean square error in this paper, how can these upper bounds be explained based on the known lower bounds (Mayekar 2020, Ramezani-Kebrya 2021, Acharya 2021)?

Prathamesh Mayekar and Himanshu Tyagi. Limits on gradient compression for stochastic optimization. In IEEE International Symposium on Information Theory (ISIT), 2020.

Ali Ramezani-Kebrya, Fartash Faghri, Ilya Markov, Vitalii Aksenov, Dan Alistarh, and Daniel M
Roy. Nuqsgd: Provably communication-efficient data-parallel sgd via nonuniform quantization.
Journal of Machine Learning Research, 22(114):1–43, 2021.

Jayadev Acharya, Clement Canonne, Prathamesh Mayekar, and Himanshu Tyagi. "Information-constrained optimization: can adaptive processing of gradients help?" Advances in Neural Information Processing Systems (NeurIPS), 2021.



------

Several closely related work have not been cited/compared:

Hantian Zhang, Jerry Li, Kaan Kara, Dan Alistarh, Ji Liu, and Ce Zhang. ZipML: Training linear models with end-to-end low precision, and a little bit of deep learning. In International Conference on Machine Learning (ICML), 2017.

Wei Wen, Cong Xu, Feng Yan, Chunpeng Wu, Yandan Wang, Yiran Chen, and Hai Li. TernGrad: Ternary gradients to reduce communication in distributed deep learning. In Advances in Neural Information Processing Systems (NeurIPS), 2017.

Fartash Faghri, Iman Tabrizian, Ilia Markov, Dan Alistarh, Daniel M. Roy, and Ali RamezaniKebrya. Adaptive gradient quantization for data-parallel SGD. In Advances in Neural Information Processing Systems (NeurIPS), 2020.

Prathamesh Mayekar and Himanshu Tyagi. Limits on gradient compression for stochastic optimization. In IEEE International Symposium on Information Theory (ISIT), 2020.


------

The expected number of bits to transmit should be added to Table 1. For example, for the same $b$, the expected number of bits to transmit for QSGD is much less than QUIC-FL since only one real value for the norm is sent precisely.


------

When $b$ and $p$ very small,  QUIC-FL becomes essentially rand-$K$ sparsification method.


**Summary Of The Paper:**

The authors study Distributed Mean Estimation problem (DME) where $n$ clients communicate a representation of a $d$-dimensional vector
to a parameter server which estimates the vectors’ mean.

This paper aimes to resolve the decoding time slowdown from these recent state of the art DME
techniques Vargaftik et al. (2021; 2022). To control quantization erorr, tthe authors send an expected $p$-fraction of the rotated coordinates exactly for some small $p$ and apply client-specific shared randomness.

**Summary Of The Review:**

Unfortunately, I think there are major technical issues in the paper. So this paper is not ready for publication.

---

> ### Author Response · Authors · 2022-11-18
> **Below, we address the raised concerns in detail.**
>
> ### Concern 1
>
> The reviewer here is just wrong. We start with the rotation, after which each coordinate is (approximately) a normal random variable, as described, and we treat it as a normal random variable throughout.  (Appendix B, for example, shows that the additional error is vanishing.)  We truncate at the quantization step, so values outside [-Tp,Tp] are sent exactly.  For those points the error is taken as  0.  (As explained in the paper, we assume 32-bit precision is sufficient.)
>
> So for example the expected square error in page 4, the integral needs only to be over the range [-Tp,Tp];  the probabilities used in the expectation are (correct) for the normal distribution. The truncation is accounted for (since the error on truncated points is 0).  The same is true for page 5.  The truncation is accounted for;  it doesn’t change the probability distribution of the coordinates, it just means that for points outside that range we leave the values uncompressed.
>
> ### Concern 2
>
> There is no contradiction that we are aware of between these works and ours.  If the reviewer could point out what they think the “contradictions” are that would be helpful.  There are other schemes with dimension-independent upper bounds on the normalized mean square error (Kashin's representation + SQ, EDEN, etc.).
>
> The reviewer unfortunately does not give enough details so that we may understand what they think the “contradiction” is.  But as one example, for the NUQSGD paper by Ramezani-Kebrya et al., Theorem 7 gives a lower bound where the quantization is given and then the adversary is allowed to pick a vector v to be quantized which would give a large quantization error (or variance in their terminology). This does not apply to QUIC-FL as we start with a random rotation, and look at the expected error;  all input vectors have the same expectation under our approach.  (i.e., the adversary cannot choose the quantized vector, just the one before the rotation.)  Again, we would be happy to consider if there is a specific contradiction the reviewer has in mind.
>
> ### Concern 3
>
> None of these works directly consider the DME problem, which is the focus of our paper.  They do not appear to actually be closely comparable papers.  We would be happy to cite any of them if there is a specific reason to do so.
>
> The closest work among these is TernGrad, which is similar to QSGD with clipping heuristics, which we do compare with (the heuristics are left out as it is not a principled extension and not generally beneficial in practice for DME).
>
> ### Concern 4
>
> This is entirely wrong. For the same communication budget, without using some variable length code, QSGD is ASYMPTOTICALLY less accurate.  The table provides the correct asymptotics for b = O(1) (a constant) as stated;  we are not sure of what the reviewer’s confusion is, other than perhaps they are considering QSGD with some entropy-based encoding can lead to an asymptotic O(1/n) NMSE.  (Note we specifically say we are not considering entropy-based encodings, which can be significantly expensive computationally;  and entropy-based additions can be added to our scheme as well.)
>
> If the reviewer means that QUIC-FL requires more bits because we send a p-fraction of the coordinates exactly, our scheme uses O(1) bits per coordinate and takes this into account (this doesn’t increase the number of bits to more than constant per coordinate as p is constant.)
>
> ### Concern 5
>
> This is wrong.  First, b is a positive integer and has to be at least 1.  The reviewer seems to be thinking of a case where b is so small that essentially no information is sent about coordinates in the range [-Tp,Tp].  To be clear, this does not happen, but if it were to happen, then the p-fraction of the most significant coordinates would be transmitted accurately. This somewhat resembles Top-K (not Rand-K), but there is a significant difference: we apply a random rotation first. The result is not similar to Rand-K in any way, and applying Top-K without rotation gives a biased result whose NMSE does not decay with the number of clients.
>
> Finally, even if this comment was true, we are not sure what the reviewer thinks the implications of that are, positive or negative.

---

### Official Review · Reviewer_GcFN · 2022-10-25

**Confidence:** 3
**Correctness:** 2
**Technical Novelty And Significance:** 2
**Empirical Novelty And Significance:** 2
**Recommendation:** 3

**Clarity, Quality, Novelty And Reproducibility:**

The clarity can be improved by polishing the presentation. The paper has some novelty in it, but the theoretical analysis is not very rigorous. Reproducibility: no code is provided but I think the results can be reproduced.

**Strength And Weaknesses:**

1. In general, the paper studies a meaningful problem of design unbiased stochastic compression for a vector. However, I have some concerns about the theoretical development. For a vector $x$, after rotation, the norm of the rotated vector $x^TR$ is fixed (equal to $||x||$). Thus, the argument that $x^TR$ is treated as iid Gaussian for designing the quantizer is not very promising. What's the convergence rate of the iid Gaussian approxmiation? I think this could be a serious issue regarding the correctness of the paper.

2. The performance of QUIC-FL only has marginal improvement over prior methods, which is not very impressive.

3. The presentation can be improved. The text on pages 3-6 is too dense. It might be helpful to add some definitions, claims, or theorems to highlight the key results.

4. In Figure 4, why there is a huge sudden drop of the test accuracy curves, including the baseline full-precision model? This seems strange and needs to be re-implemented.

5. Quantization for Gaussian distribution is a classical topic in literature. There should be a related work section on quantization of Gaussian (and other) random variables. The novelty of the proposed quantization method is not very clear.

**Summary Of The Paper:**

The paper proposes an unbiased stochastic quantization method called QUIC-FL for rotation-based mean estimation, in order to reduce the communication by transmitting full-precision vectors. The method uses a 'shared randomness' approach which reduces the computation cost at the server to recover the compressed signals. The theoretical foundation is based on a Gaussian approximation argument, and the algorithm is realized by relaxing a continuous problem to a discrete problem. Detailed algorithm design and error analysis are presented. Experiments on simulated and real FL datasets shows the effectiveness of the proposed QUIC-FL.

**Summary Of The Review:**

The paper presents meaningful results on unbiased stochastic quantization. However, the analytical assumption on the Gaussian approximation, which is the building block of the paper, might be problematic. Thus, this issue needs to be addressed properly. Besides, the empirical gain is not very significant.

---

> ### Author Response · Authors · 2022-11-18
> **Below, we address the raised concerns in detail.**
>
> ### Concern 1
>
> It is well known that each coordinate after a random rotation is distributed as a Gaussian, conditioned on the norm of the entire rotated vector being $||x||$.  The reviewer themself seemed to know this.  This is the only dependence;  it is clear that as the dimension gets large this is not significant.  Further, to the extent it is significant in deriving theoretical results, this is handled by the proofs in the appendices, where we do not assume independence.  (e.g.  See Appendix B, which specifically shows that the difference in vNMSE from the rotated vector and assuming a normally distributed coordinate is vanishing.)
>
> We invite the reviewer to take a look at the papers (e.g., DRIVE, EDEN) we reference that already tackle this question and show that this approximation is perfectly valid. Indeed, in the paper we specifically cite DRIVE for this:  “QUIC-FL also leverages random rotations, and uses the observation that after rotation the coordinates’ distribution approaches d i.i.d. normal variables, $\mathcal{N}(0,|||x||_2^2/d)$ [5].”
>
> ### Concern 2
>
>  As we make clear in the paper, the improvement we aim for in QUIC-FL is to avoid having to invert the rotation separately for each client at the server, which gives an asymptotic improvement in the decompression time.  We show that this can be accomplished with minimal change in accuracy from previous (best-in-the-literature) schemes that do invert for each client.  We do not consider this practical and asymptotic improvement in the decoding speed as marginal. Perhaps the reviewer could explain what they mean by “not very impressive”, which seems both subjective and impossible to argue.
>
> ### Concern 3
>
> For space reasons, much of these formalizations appear in the appendices. If the reviewer has any specific suggestions of things they feel it is important to put in the body of the paper, that might be helpful.
>
> ### Concern 4
>
> There are some fluctuations in the zoomed-in plots. This is consistent in federated optimization experiments at such a zoom-level.  As such, we are not clear about what sudden drops the reviewer is referring to.
>
> ### Concern 5
>
> Our approach combines the use of truncating and shared randomness, and then uses a solver for the specific distribution to obtain an unbiased quantization.  Other approaches that we are aware of do not have our combination of properties.  The Lloyd-max quantizer, for example, is biased.  Subtractive dithering is distribution-agnostic.  Our solver sometimes causes us to quantize to something other than one of the two nearest quantization levels, in order to realize unbiasedness, which is an unusual (although we are not sure it is unique) feature.
>
> We are happy to add a discussion of this in the paper.
>
> If the reviewer is aware of a work in the literature that provides an **unbiased** quantization method optimized for the normal (or truncated normal) distribution, we will happily mention and compare to it.

---

### Official Review · Reviewer_Whkw · 2022-11-03

**Confidence:** 2
**Correctness:** 3
**Technical Novelty And Significance:** 2
**Empirical Novelty And Significance:** 3
**Recommendation:** 3

**Clarity, Quality, Novelty And Reproducibility:**

Clarity:
This paper is unclear and should be refactored to be more readable.

Reproducibility:
The code is provided, but I haven't checked the details.

**Strength And Weaknesses:**

Strength:
1. Comprehensive experiments over different datasets are conducted, and QUIC-FL outperforms conventional baselines and is comparable with EDEN. Different ablation studies are provided, such as different bit budgets, number of clients, and dimensions.

Concerns:
1. This paper is hard to follow. Besides, the math symbols should be highly reduced and unified. In this case, I can not understand exactly what the algorithm is doing and why it works. Moreover, I would suggest Alg. 1 be better described with more details, then more extended descriptions to be given based on Alg. 1. It gives me a feeling of the current Alg. 1 is only an unreadable sketch. This work should also be more centralized and have a better structure.
2. In some figures, such as CIFAR-10 in Fig. 5, both the training accuracy and the testing accuracy for Float32 is lower than EDEN and the proposed QUIC-FL. I doubt the results here. More analysis should be provided as well.
3. It's unclear what is the convergence speed of QUIC-FL. Only the vNMSE upper bound is provided, but I assume it's insufficient. This paper is called \textit{Quick} compression; I'm not quite sure whether the convergence speed of QUIC-FL is promising. Besides, are the operators of QUIC-FL computationally heavy?

**Summary Of The Paper:**

This work introduces an unbiased compression pipeline for Distributed Mean Estimation (DME) and achieves good performance over several typical datasets compared with competitive baselines.

**Summary Of The Review:**

I'm highly confused with the writing and structure of this paper—also have some doubts about the results. The details are provided above.

I would like to receive some comments from other reviewers and feedback from the author(s).

---

> ### Author Response · Authors · 2022-11-18
> **Below, we address the raised concerns in detail.**
>
> ### Concern 1
> We used different math symbols to denote different quantities. Yes, the algorithm is not simple and requires appropriate notation. However, given that the symbols are well-understood, Algorithm 1 is not a sketch but a precise description of the solution.
>
> If the reviewer would like to point out specific  math symbols that they thought were redundant or point to symbols that they thought were not clearly defined we could address specific comments.
>
> ### Concern 2
> In non-convex optimization and especially DNNs, it is not surprising that adding some noise during the training procedure may improve the resulting accuracy (i.e., generalization properties). This observation is not new and has been observed by previous papers (such as DRIVE, e.g., in Figure 2). Importantly, and as one might expect, the gap between EDEN, QUIC-FL, and the baseline (full precision) is not significant or persistent. Overall, all three have comparable accuracy in the presented experiments.
>
> ### Concern 3
> QUIC-FL produces unbiased estimates with a constant NMSE. Therefore, it’s convergence speed essentially matches the vanilla SGD (e.g., Karimireddy et al., ICML 2019). Also, it is known that the provable convex convergence rates for compressed SGD have a linear dependence on the NMSE (Bubeck (2015), Theorem 6.3).
>
> As noted in DRIVE [see before Theorem 5 in the arXiv version]:  an O(1) vNMSE for the unbiased estimate guarantees (e.g., see  arXiv:2002.12410) that distributed SGD converges at the same asymptotic rate as without compression.
>
> The operators of QUIC-FL are not computationally heavy (particularly when we use the randomized Hadamard transform, as described).  They are on par or less than other rotation-based algorithms (see Figure 2), and in particular the reduction in communication outweighs the cost of additional computation, and (as noted above) generally outweighs any costs related to convergence speed;  this is why compression is studied in this context.

---

### Author Response · Authors · 2022-11-18
**The level of the reviews is disappointing and below our expectations for ICLR.**

Overall, we are deeply disappointed by the poor level of the reviews. (They are excellent examples of How Not to Review a Paper, as described by https://sigmodrecord.org/publications/sigmodRecord/0812/p100.open.cormode.pdf).

In some cases, it is evident that the paper was not understood. In other cases, the reviewers make generic, non-specific criticisms (such as “the presentation can be improved”) without details. The most notable suggested addition (on past related work related to quantizing Gaussians) can be easily added, but (we feel) somewhat tangential to our work.  Otherwise, we’re afraid we reject the reviews, which are below our expectations for ICLR.

---

### Decision · Program_Chairs · 2023-01-20

**Decision:**

Reject

**Justification For Why Not Higher Score:**

All reviewers gave this paper score 3. I can't make any other decision without reading the paper myself (I did not) or without further reviews (I do not have more reviews available).

**Justification For Why Not Lower Score:**

This is the lowest score already.

**Metareview: Summary, Strengths And Weaknesses:**

All reviewers suggested a rejection (score 3), and the author response did not change their mind towards a favorable outcome. Hence, I have no other recourse but to recommend a rejection of the paper. There is no need to summarize the strengths and weaknesses in this very clear-cut case. Thanks to the authors for submitting their work.

AC